# TRIM27 mediates STAT3 activation at retromer-positive structures to promote colitis and colitis-associated carcinogenesis

Hong-Xia Zhang[1,2], Zhi-Sheng Xu[3,4], Hen Lin[1], Mi Li[1], Tian Xia[1], Kaisa Cui[1], Su-Yun Wang[4], Youjun Li[1], Hong-Bing Shu [ID] [1,2] & Yan-Yi Wang[4]

STAT3 is a transcription factor that plays central roles in various physiological processes and its deregulation results in serious diseases including cancer. The mechanisms on how STAT3 activity is regulated remains enigmatic. Here we identify TRIM27 as a positive regulator of Il-6-induced STAT3 activation and downstream gene expression. TRIM27 localizes to retromer-positive punctate structures and serves as a critical link for recruiting gp130, JAK1, and STAT3 to and subsequent phosphorylation of STAT3 at the retromer-positive structures. Over-expression of TRIM27 promotes cancer cell growth in vitro and tumor growth in nude mice, whereas knockdown of TRIM27 has opposite effects. Deficiency of TRIM27 significantly impairs dextran sulfate sodium (DSS)-induced STAT3 activation, inflammatory cytokine expression and colitis as well as azoxymethane (AOM)/DSS-induced colitis-associated cancer in mice. These findings reveal a retromer-dependent mechanism for regulation of STAT3 activation, inflammation, and inflammation-associated cancer development.

---

[1] College of Life Sciences, Wuhan University, 430072 Wuhan, China. [2] Medical Research Institute, School of Medicine, Wuhan University, 430071 Wuhan, China. [3] The Joint Center of Translational Precision Medicine, Guangzhou Institute of Pediatrics, Guangzhou Women and Children's Medical Center, 510623 Guangzhou, China. [4] Wuhan Institute of Virology, Chinese Academy of Sciences, 430071 Wuhan, China. These authors contributed equally: Hong-Xia Zhang, Zhi-Sheng Xu. Correspondence and requests for materials should be addressed to H.-B.S. (email: shuh@whu.edu.cn) or to Y.-Y.W. (email: wangyy@wh.iov.cn)

Chronic inflammation plays important roles in pathogenesis of many types of malignancies from leukemia to various solid tumors. Recent studies have highlighted critical roles of the master transcription factors nuclear factor κB (NF-κB) and signal transducer and activator of transcription 3 (STAT3) in linking inflammation to cancer development[1,2]. Specific ablation of the IκB kinase β (IKKβ) or deficiency of the interleukin-6

(IL-6)-STAT3 axis inhibits colitis-associated cancer (CAC) development in mouse models[3,4]. In addition to CAC, the NF-κB-IL-6-STAT3 axis is also closely linked to inflammatory processes during development of liver, lung, and pancreatic cancers[5–8]. Although the critical roles of STAT3 in tumorigenesis are evident, the mechanisms on how activation of STAT3 is regulated remains enigmatic.

Activation of STAT3 is tightly regulated during various physiological processes, such as cell proliferation, survival, and differentiation. Aberrant and persistent activation of STAT3 has been found in various types of cancers[9]. Despite several cytokines or growth factors such as IL-11, oncostatin M (OSM), leukemia inhibitory factor (LIF), epidermal growth factor (EGF), and hepatocyte growth factor (HGF) can activate STAT3, IL-6 is the most important STAT3 activator in various tumors[10]. Binding of IL-6 to its receptor IL-6Rα/gp130 at plasma membrane leads to activation of JAKs that are constitutively associated with gp130. Activated JAKs then mediate phosphorylation of gp130, leading to the recruitment of cytosolic STAT3 and its phosphorylation at Y705. Phosphorylated STAT3 then dimerizes and translocates into the nucleus to induce transcription of a set of downstream genes, which play crucial roles in promoting tumor cell proliferation and survival, tumor invasion, angiogenesis, and immunosuppression[11].

Endocytosis is a cellular process by which cell surface components and extracellular molecules are internalized into cellular vesicles[12]. There is growing evidence that the endosomal system not only serves as a conduit for the degradation or recycling of cell surface receptors, but also functions as an essential site of signal transduction. It has been reported that receptor-mediated endocytosis is required for STAT3 activation[13,14]. In addition, it has been shown that phosphorylated STAT3 induced by EGF and HGF co-localizes with endosomal vesicles in the cytoplasm[13,14]. However, phosphorylated STAT3 induced by IL-6 is sequestered into unidentified membrane vesicles which are distinct from the endosomal vesicles at where the EGF- and HGF-induced phosphorylated STAT3 is localized and are negative for various endosomal markers such as EEA1, Rab5, Rab7, and Rab11 among others[15]. These studies imply that there exists remarkable differences in STAT3 activation in response to different stimuli and the detailed mechanisms on how vesicular trafficking regulates STAT3 activation remain elusive.

The TRIM family of proteins is characterized by its tripartite motif (TRIM), which is composed of a RING finger domain, one or two B-box domains, a coiled-coil domain[16]. The TRIM family proteins are involved in various physiological processes and their alterations result in diverse pathological conditions such as developmental disorders, immunological diseases, and tumorigenesis[16,17]. *TRIM27* (also called *RFP*) was originally identified as a gene involved in oncogenic rearrangements with the *RET* proto-oncogene[18]. It has been shown that TRIM27 is highly expressed in various cancers including breast, endometrial, ovarian, lung, and colon cancers[19–23]. TRIM27 is a multifunctional protein which is involved in cell proliferation, transcriptional repression, negative regulation of NF-κB activation, apoptosis, and innate immune response[24–31]. In a screen for proteins that regulate STAT3 activity, we identify TRIM27 as an important mediator of IL-6-induced STAT3 activation. Interestingly, we find that TRIM27 was localized at retromer-positive structures. Following IL-6 stimulation, the retromer-localized TRIM27 recruited JAK1 and STAT3, leading to STAT3 phosphorylation and induction of downstream effector genes. Furthermore, deficiency of TRIM27 significantly impairs STAT3 activation, suppresses dextran sulfate sodium (DSS)-induced colitis, and azoxymethane (AOM)/DSS-induced CAC development in mice. Our findings reveal a retromer-dependent mechanism for STAT3 activation and inflammation-associated cancer development.

## Results

**TRIM27 mediates STAT3 activation.** To identify candidate proteins that regulate STAT3 activity, we screened ~13,000 independent human and murine cDNA expression plasmids by reporter assays[32]. These screens identified TRIM27 as a protein that could modulate STAT3 activation. As shown in Fig. 1a, overexpression of TRIM27 activated STAT3 and potentiated IL-6-induced STAT3 activation in a dose-dependent manner. In similar experiments, TRIM27 did not activate IFN-β-induced STAT1/2 activation (Fig. 1b) and marginally enhanced IFN-γ-induced STAT1 activation (Supplementary Figure 1a), suggesting that TRIM27 modulates STAT3 activity specifically. Consistently, overexpression of TRIM27 potentiated IL-6-induced transcription of downstream effector genes such as *SOCS3* and *FOS* (Fig. 1c) as well as STAT3 phosphorylation at Y705 (Fig. 1d), which is a hallmark of STAT3 activation. We next investigated whether endogenous TRIM27 was involved in STAT3 activation. Knockdown of TRIM27 by two independent RNAi plasmids significantly inhibited IL-6-induced transcription of *SOCS3*, *IL-6*, and *FOS* and STAT3 activation (Fig. 1e, f) but not IFN-β-induced STAT1/2 activation in HeLa cells (Fig. 1g). Consistently, knockdown of TRIM27 markedly inhibited IL-6-induced phosphorylation of STAT3 at Y705 (Fig. 1h). Similar results were also obtained in the colonic epithelial RKO and HT29 cells (Supplementary Figure 1b–h). These results suggest that TRIM27 mediates IL-6-induced STAT3 activation.

**Fig. 1** TRIM27 mediates STAT3 activation. **a** Effects of TRIM27 on IL-6-induced STAT3 activation. **b** Effects of TRIM27 on IFN-β-induced activation of STAT1/2. HEK293 and HeLa cells ($5 \times 10^4$) were transfected with indicated reporter (10 ng) and increased amounts of TRIM27 expression plasmids. Twenty hours after transfection, cells were treated with IL-6 (20 ng/mL) in **a** and IFN-β (20 ng/mL) in **b**, respectively, or left untreated for 10 h in serum-free DMEM before luciferase assays were performed. **c**, **d** Effects of TRIM27 on IL-6-induced transcription of *SOCS3* and *FOS* genes and STAT3 phosphorylation. HEK293 cells ($2 \times 10^5$) were transfected with an empty vector or TRIM27 expression plasmid (0.1 μg). Twenty hours after transfection, cells were starved with serum-free DMEM overnight followed by IL-6 treatment (20 ng/mL) for 1 h before qPCR experiments (**c**) and for the indicated times before immunoblotting analysis (**d**). **e** Effects of TRIM27 knockdown on IL-6-induced transcription of *SOCS3*, *IL-6*, and *FOS* genes. **f** Effects of TRIM27 knockdown on IL-6-induced activation of STAT3 reporter. **g** Effects of TRIM27 knockdown on IFN-β-induced activation of STAT1/2 reporter. **h** Effects of TRIM27 knockdown on IL-6-induced STAT3 phosphorylation. The control and TRIM27-RNAi HeLa cells ($2 \times 10^5$) were subjected to qPCR experiments (**e**), luciferase reporter assays (**f**, **g**), and immunoblotting analysis (**h**) as described in **a–d**. **i** Effects of TRIM27 deficiency on IL-6-induced transcription of *Socs3*, *Jun*, and *Fos* genes in BMDMs. **j** Effects of TRIM27 deficiency on IL-6-induced STAT3 phosphorylation in BMDMs. *Trim27*$^{+/+}$ and *Trim27*$^{-/-}$ BMDMs ($2 \times 10^5$) were starved overnight and stimulated with IL-6 (15 ng/mL) for the indicated times before qPCR experiments (**i**) and immunoblotting analysis (**j**). **k** Effects of TRIM27 deficiency on IL-6-induced transcription of *Socs3* and *Il6* genes in IECs. **l** Effects of TRIM27 deficiency on IL-6-induced STAT3 phosphorylation in IECs. *Trim27*$^{+/+}$ and *Trim27*$^{-/-}$ IECs ($2 \times 10^5$) were stimulated with IL-6 (50 ng/mL) for the indicated times before qPCR experiments (**k**) and immunoblotting analysis (**l**). Data are representative of three experiments with similar results. Graphs show mean ± SD; $n = 3$. *$P < 0.05$, **$P < 0.01$, ***$P < 0.001$, unpaired $t$ test

To further explore the functions of TRIM27 in vivo, TRIM27-deficient mice were generated using the CRISPR/Cas9 strategy (Supplementary Figure 2a). Deficiency of TRIM27 in the knockout mice was confirmed by genotyping and immunoblotting analysis (Supplementary Figure 2b). We prepared murine bone-marrow-derived macrophages (BMDMs) from the knockout mice and found that TRIM27 deficiency markedly inhibited IL-6-induced transcription of downstream genes including *Socs3*, *Jun*, and *Fos* (Fig. 1i). TRIM27 deficiency also markedly inhibited IL-6-induced phosphorylation of STAT3 Y705 but not JAK1 Y1022/1023

(Fig. 1j). Similarly, TRIM27 deficiency also inhibited IL-6-induced transcription of downstream genes and STAT3 Y705 phosphorylation in murine primary intestinal epithelial cells (IECs) (Fig. 1k, l) and hepatocytes (Supplementary Figure 2c, d). These results confirm that TRIM27 is important for IL-6-mediated STAT3 activation in divergent types of cells.

We next determined whether TRIM27 is involved in STAT3 activation triggered by other stimuli. Overexpression of TRIM27 potentiated STAT3 activation triggered by OSM, another member of IL-6 family cytokines (Supplementary Figures 3a). Knockdown

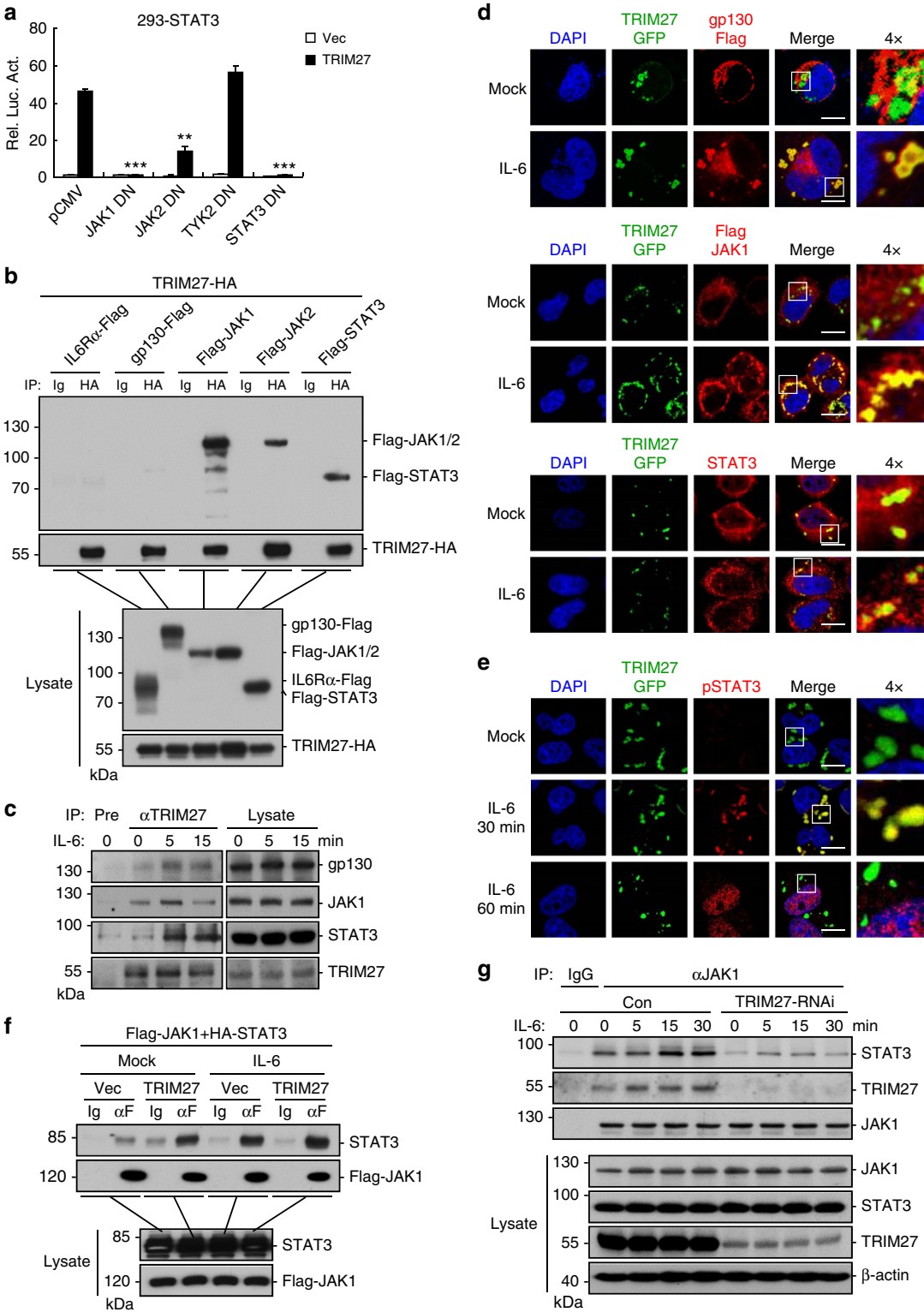

of TRIM27 in HeLa cells inhibited OSM-induced transcription of *SOCS3*, *IL6*, and *FOS* genes (Supplementary Figure 3b) and STAT3 Y705 phosphorylation (Supplementary Figure 3c). TRIM27 deficiency in primary mouse cells also inhibited OSM-induced transcription of downstream genes (Supplementary Figure 3d) and STAT3 Y705 but not JAK1 Y1022/1023 phosphorylation (Supplementary Figure 3e).

**TRIM27 is essential for JAK1–STAT3 complex formation**. We next investigated the molecular mechanisms by which TRIM27 regulates STAT3 activity. Overexpression of dominant-negative mutants of STAT3 and its upstream components, including JAK1 (K908A), JAK2(K882A), TYK2(K930A), and STAT3(Y705F) effectively inhibited IL-6-induced STAT3 activation in reporter assays (Supplementary Figure 4a). However, TRIM27-mediated activation of STAT3 was abolished by JAK1(K908A) and STAT3 (Y705F) and partially inhibited by JAK2(K882A) but not TYK2 (K930A) (Fig. 2a). Coimmunoprecipitation experiments indicated that TRIM27 interacted with JAK1, JAK2, and STAT3, and the interaction between TRIM27 and JAK1 was much stronger than that of TRIM27 and JAK2 (Fig. 2b). In addition, endogenous TRIM27 was associated with gp130, JAK1, and STAT3, and their associations were enhanced upon IL-6 stimulation (Fig. 2c). Confocal microscopy indicated that TRIM27 existed in discrete cytoplasmic punctate structures and was co-localized with a fraction of gp130, JAK1, and STAT3 after IL-6 stimulation for 30 min (Fig. 2d). Notably, phosphorylated STAT3 was well co-localized with TRIM27 in punctate structures in the cytoplasm at 30 min post IL-6 stimulation and then translocated into the nucleus at 60 min post IL-6 stimulation (Fig. 2e). These results suggest TRIM27 forms complex with JAK1 and STAT3 in cytoplasmic punctate structures following IL-6 stimulation.

Previously, it has been reported that TRIM27 has E3 ubiquitin ligase activity[28,33]. Therefore, we determined whether TRIM27-mediated STAT3 activation is dependent on its E3 ubiquitin ligase activity. We found that the enzymatic inactive mutants TRIM27(C/S) and TRIM27ΔRing were able to potentiate IL-6-induced STAT3 activation as well as wild-type TRIM27 in both HEK293 and HeLa cells (Supplementary Figure 4b). In addition, TRIM27 had no obvious effects on ubiquitination of JAK1 and STAT3 (Supplementary Figure 4c, d). These results suggest that the E3 ubiquitin ligase activity of TRIM27 is dispensable for its ability to mediate STAT3 activation.

Since TRIM27 is associated with JAK1 and required for full phosphorylation of STAT3 after IL-6 stimulation, we investigated whether TRIM27 could affect complex formation of JAK1 and STAT3. Coimmunoprecipitation indicated that overexpression of TRIM27 promoted the association between JAK1 and STAT3 (Fig. 2f), while its knockdown markedly reduced IL-6-induced JAK1–STAT3 association (Fig. 2g). These results suggest that TRIM27 mediates the association between JAK1 and STAT3 after IL-6 stimulation.

**Retromer acts as a platform for IL-6-induced STAT3 activation**. Previously, it has been reported that TRIM27 is localized to retromer-containing endosomes[33]. Retromer is a highly conserved heteropentameric complex important for recycling proteins from the endosomes to trans-Golgi network or plasma membrane. The mammalian retromer complex comprises a core cargo recognition trimer composed of VPS26, VPS29, and VPS35, and a sorting nexin heterodimer composed of SNX1 or SNX2 with SNX5, SNX6, or SNX27[34]. To confirm whether the TRIM27-containing cytoplasmic punctate structures observed in our earlier experiments (Fig. 2d, e) are indeed retromer-containing structures, we further performed confocal microscopic study. We found that TRIM27 specifically colocalized with the retromer component VPS26 but not other endosomal markers such as EEA1 and RhoB (Fig. 3a). This led us to investigate whether retromer-containing punctate structures are involved in IL-6-STAT3 signaling. Confocal microscopy indicated that phosphorylated STAT3 induced by IL-6 was well colocalized with the retromer marker VPS26 but not found in EEA1-, Rab5-, and RhoB-positive endosomes or LAMP1-positive lysosomes (Fig. 3b). In addition, phosphorylated STAT3 and TRIM27 were well colocalized with VPS26 (Fig. 3c). Coimmunoprecipitation experiments indicated that the retromer component VPS35 could interact with JAK1, JAK2, and STAT3, but not IL6-Rα or gp130 in overexpression systems (Fig. 3d). It has been reported that IL-6 stimulation induces endocytosis of the IL-6Rα/gp130 receptor complexes, we therefore examined whether endocytosis was required for IL-6-induced STAT3 activation. As shown in Supplementary Figure 4e, f, IL-6-induced activation of STAT3 as well as its phosphorylation at Y705 was inhibited by treatment with the dynamin inhibitor dynasore[35], indicating clathrin-dependent endocytosis of the IL-6Rα/gp130 receptor complexes is required for IL-6-induced STAT3 activation. This led us to investigate whether VPS35 interacted with the signaling components following IL-6 stimulation. As the available antibodies against VPS35 could not enrich endogenous VPS35, we generated a HeLa cell line in which relative low level of Flag-tagged VPS35 was expressed. As shown in Fig. 3e, Flag-VPS35 associated with endogenous gp130, JAK1, and STAT3 upon IL-6 stimulation,

**Fig. 2** TRIM27 promotes JAK1–STAT3 complex formation. **a** Effects of various dominant-negative mutants on TRIM27-mediated STAT3 activation. HEK293 cells ($5 \times 10^4$) were transfected with STAT3 reporter (10 ng), TRIM27 and the indicated mutant plasmids (0.1 μg each) for 24 h before luciferase assays. **b** TRIM27 interacts with JAK1, JAK2, and STAT3 in mammalian overexpression system. HEK293 cells ($2 \times 10^6$) were transfected with the indicated plasmids for 24 h. Coimmunoprecipitation and immunoblot analysis were performed with the indicated antibodies. **c** Endogenous TRIM27 is associated with gp130, JAK1, and STAT3. HeLa cells ($2 \times 10^7$) were starved overnight and then treated with IL-6 (50 ng/mL) or left untreated for the indicated times. Coimmunoprecipitation and immunoblot analysis were performed with the indicated antibodies. **d** Colocalization of TRIM27, gp130, JAK1, and STAT3. HeLa cells ($1 \times 10^5$) were transfected with GFP-tagged TRIM27 (0.2 μg) and Flag-tagged gp130 (0.1 μg) or Flag-tagged JAK1 (0.1 μg) expression plasmids. Twenty hours after transfection, cells were starved overnight followed by stimulation with IL-6 (50 ng/mL) for 30 min. Immunostaining was performed with anti-Flag and anti-STAT3. Scale bars, 10 μm. **e** Colocalization of TRIM27 and pY705-STAT3. The experiments were performed similarly performed as in **d**, except that antibody against pY705-STAT3 was used. Scale bars, 10 μm. **f** Overexpression of TRIM27 promotes JAK1–STAT3 interaction. HEK293 cells ($2 \times 10^6$) were transfected with Flag-tagged JAK1 (4 μg) and HA-tagged STAT3 (3 μg) plasmids together with empty or TRIM27 expression plasmid (2 μg). Twenty-four hours after transfection, cells were starved overnight followed by stimulation with IL-6 (50 ng/mL) for 30 min. Coimmunoprecipitation and immunoblot analysis were performed with the indicated antibodies. **g** Knockdown of TRIM27 impairs JAK1–STAT3 interaction. The control and TRIM27-RNAi HeLa cells ($2 \times 10^7$) were starved overnight followed by stimulation with IL-6 (50 ng/mL) for the indicated times. Coimmunoprecipitation and immunoblot analysis were performed with the indicated antibodies. Data are representative of three experiments with similar results. Graphs show mean ± SD; $n = 3$. *$P < 0.05$, **$P < 0.01$, ***$P < 0.001$, unpaired $t$ test

whereas Flag-VPS35 constitutively interacted with endogenous VPS26 and TRIM27. Confocal microscopy indicated that gp130 colocalized with VPS26 and TRIM27 following IL-6 stimulation (Fig. 3f). These results indicate gp130 underwent endocytosis and migrated to the retromer following IL-6 stimulation. Knockdown of VPS35 inhibited IL-6-induced STAT3 phosphorylation (Fig. 3g). In similar experiments, knockdown of VPS35 prolonged IFN-α-induced STAT1 phosphorylation (Fig. 3g), which is consistent with a previous report[36]. These results suggest that retromer is essential for IL-6-induced phosphorylation of STAT3.

We further investigated whether the retromer acts as a platform for assembly of gp130–JAK1–STAT3 complex. Coimmunoprecipitation experiments indicated that IL-6 stimulation increased associations of JAK1 with the retromer component VPS35, as well as with gp130, STAT3, and TRIM27, whereas knockdown of VPS35 decreased these associations (Fig. 4a). Knockdown of TRIM27 also impaired the associations of JAK1 with VPS35, VPS26, or STAT3 (Fig. 4b). Moreover, knockdown of TRIM27 impaired the associations of VPS26 with gp130, JAK1, or STAT3, but had no effects on the association between VPS35 and VPS26 (Fig. 4c). Consistently, knockdown of each of the

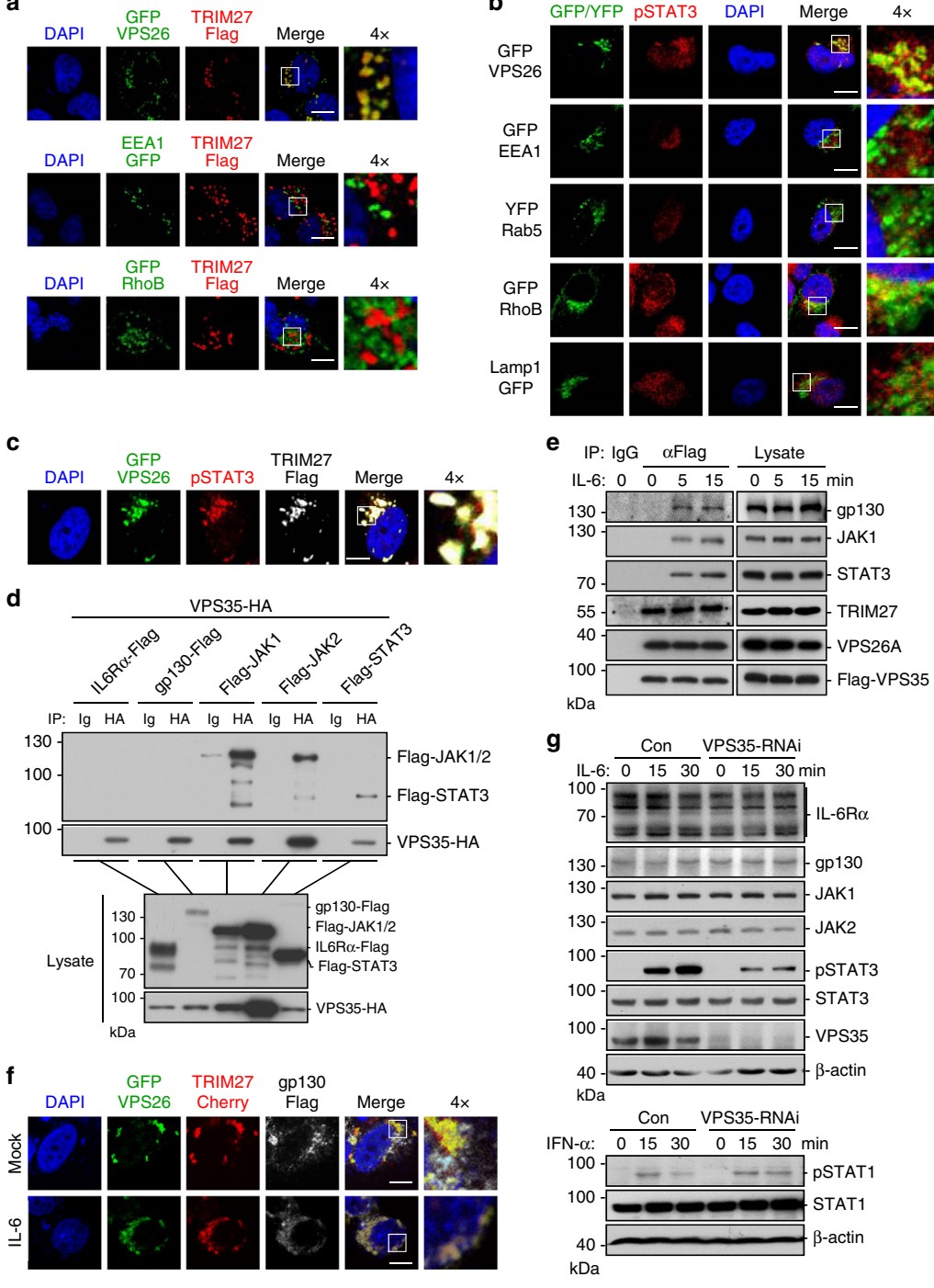

retromer components VPS26, VPS29, or VPS35 significantly inhibited IL-6-induced transcription of *SOCS3* and *FOS* genes (Fig. 4d). Collectively, these results suggest that the retromer-positive structure acts as a platform for TRIM27-mediated assembly of gp130–JAK1–STAT3 complexes and subsequent phosphorylation of STAT3 following IL-6 stimulation.

**TRIM27 plays a role in cell malignancy.** Since TRIM27 modulates IL-6-induced STAT3 activation, which has been reported to play important roles in tumorigenesis[10], we attempted to investigate the roles of TRIM27 in tumorigenesis. Overexpression of TRIM27 markedly promoted anchorage-independent growth of RKO, HEK293, and HeLa cells (Fig. 5a and Supplementary Figure 5a, b) in vitro and tumor growth in nude mice (Fig. 5b and Supplementary Figure 5c), whereas knockdown of TRIM27 had opposite effects (Fig. 5c, d and Supplementary Figure 5d, e, f). These results suggest that TRIM27 plays an important role in tumorigenesis.

**Deficiency of TRIM27 attenuates DSS-induced colitis.** Because STAT3 plays important roles in inflammation-associated tumorigenesis, we next examined the effects of TRIM27 deficiency on DSS-induced acute colitis. Mice were treated with 3% DSS in drinking water over a 10-day period to induce acute colitis. We found that *Trim27*$^{-/-}$ mice displayed attenuated colitis, as determined by less weight loss (Fig. 6a), higher survival rate (Fig. 6b), and reduced colon shortening (Fig. 6c) compared to their wild-type littermates. Histopathological analysis revealed that the colonic mucosa of *Trim27*$^{-/-}$ mice was more intact without apparent loss of crypt structures and mucosal ulceration and the colonic tissues of *Trim27*$^{-/-}$ mice had less infiltration of inflammatory cells in comparison to that of their wild-type littermates after DSS treatment (Fig. 6d), which was also reflected in the pathological assessment of colitis severity scores (Fig. 6e). These results suggest that TRIM27 deficiency reduces the severity of DSS-induced colitis.

Consistent with above phenotypic consequences, both of phosphorylated STAT3 and p65 were decreased in the DSS-treated colonic lysates of *Trim27*$^{-/-}$ mice in comparison with their wild-type littermates (Fig. 6f). The mRNA levels of inflammatory cytokine genes including *Il6*, *Tnfa*, and *Il17a* as well as the STAT3 target gene *Socs3* were decreased in colonic lysates of *Trim27*$^{-/-}$ mice in comparison to their wild-type littermates (Fig. 6g). Consistently, the levels of TNF-α and IL-6 in either the serum or colonic mucosa were dramatically decreased in *Trim27*$^{-/-}$ mice in comparison to their wild-type littermates (Fig. 6h). Similarly, other inflammatory cytokines IL-1β and

IL-17A released from the colonic mucosa were also decreased in *Trim27*$^{-/-}$ mice (Fig. 6h). Collectively, these results suggest that TRIM27 deficiency reduces DSS-induced inflammatory responses.

**Deficiency of TRIM27 in hematopoietic cells is responsible for attenuation of DSS-induced colitis.** To determine which cell types that are crucial for attenuated DSS-induced colitis in TRIM27-deficient mice, four groups of reciprocal bone marrow chimeric mice were generated by adoptively transferring bone marrow to lethally irradiated recipients. Eight weeks after bone marrow transplantation, these mice were subjected to DSS challenge. As expected, TRIM27-deficient mice transplanted with hematopoietic cells from TRIM27-deficient mice (KO > KO) were less susceptible to DSS-induced colitis than wild-type mice receiving hematopoietic cells from wild-type mice (WT > WT) as determined by less body weight loss, higher survival rate, reduced colon shortening, less inflammatory cells infiltration, lower histological score, decreased level of phosphorylated STAT3, and p65 and lower expression of inflammatory cytokines (Fig. 7a–h). However, TRIM27-deficient mice transplanted with hematopoietic cells from wild-type mice (WT > KO) developed colitis as severe as that of WT > WT mice (Fig. 7a–h). Conversely, wild-type mice transplanted with hematopoietic cells from TRIM27-deficient mice (KO > WT) showed decreased susceptibility to colitis as KO > KO mice (Fig. 7a–h). These results indicate TRIM27-deficient hematopoietic cells are responsible for attenuation of DSS-induced colitis.

**TRIM27 deficiency suppresses AOM/DSS-induced CAC.** We next investigated the roles of TRIM27 in inflammation-associated cancer development by utilizing the AOM/DSS model[3]. The mice were injected with AOM followed by three rounds of 2.5% DSS exposure to trigger CAC (Fig. 8a). This procedure did not cause notable differences on the body weights, production of proinflammatory cytokines, and type I IFNs between wild-type and TRIM27-deficient mice (Supplementary Figure 6). The results indicated that TRIM27-deficient mice developed less and smaller colon tumors compared with their wild-type littermates (Fig. 8b). In addition, most of the colon adenomas in *Trim27*$^{-/-}$ mice were low-grade dysplasia, while the adenomas in their wild-type littermates were high-grade dysplasia and infiltrated with more inflammatory cells (Fig. 8c). Consistently, there was a significant decrease in proliferation rates in the colon cancers of *Trim27*$^{-/-}$ mice as determined by Ki-67 nuclear staining (Fig. 8c). Immunohistochemical analysis indicated that there were less phosphorylated STAT3-positive cells in colon tumors of TRIM27-

**Fig. 3** The retromer acts as a platform for IL-6-induced STAT3 activation. **a** TRIM27 localizes to retromer-containing punctate structures. HeLa cells (1 × 10$^5$) were transfected with Flag-tagged TRIM27 (0.2 μg) and GFP-tagged VPS26, EEA1, and RhoB plasmids (0.1 μg), respectively. Twenty-four hours after transfection, immunostaining was performed using anti-Flag antibody. **b** pY705-STAT3 localizes to retromer-containing punctate structures. HeLa cells (1 × 10$^5$) were transfected with GFP-tagged VPS26, EEA1, RhoB and Lamp1, and YFP-tagged Rab5 plasmids (0.1 μg), respectively. Twenty hours after transfection, cells were starved overnight followed by stimulation with IL-6 (50 ng/mL) for 30 min. Immunostaining was performed using antibody against pY705-STAT3. **c** Colocalization of TRIM27 and pY705-STAT3 to retromer-containing punctate structures. HeLa cells (1 × 10$^5$) were transfected with Flag-tagged TRIM27 (0.2 μg) and GFP-tagged VPS26 (0.1 μg) plasmids. Twenty-four hours after transfection, immunostaining was performed using antibodies against Flag tag and pY705-STAT3. **d** VPS35 interacts with JAK1, JAK2, and STAT3 in mammalian overexpression system. HEK293 cells (2 × 10$^6$) were transfected with the indicated plasmids for 24 h. Coimmunoprecipitation and immunoblot analysis were performed with the indicated antibodies. **e** VPS35 is associated with endogenous gp130, JAK1, STAT3, TRIM27, and VPS26. HeLa cells (2 × 10$^7$) stably expressing low-level Flag-tagged VPS35 were starved overnight and then treated with IL-6 (50 ng/mL) or left untreated for the indicated times. Coimmunoprecipitation and immunoblot analysis were performed with the indicated antibodies. **f** Colocalization of gp130 and TRIM27 to retromer-containing punctate structures. HeLa cells (1 × 10$^5$) were transfected with GFP-tagged VPS26 (0.1 μg), Cherry-tagged TRIM27 (0.2 μg) and Flag-tagged gp130 (0.1 μg) plasmids. Twenty-four hours after transfection, immunostaining was performed using anti-Flag antibody. **g** Effects of VPS35 knockdown on IL-6-induced STAT3 phosphorylation and IFN-α-induced STAT1 phosphorylation. The control and VPS35-RNAi HeLa cells (2 × 10$^5$) were stimulated with IL-6 (20 ng/mL) or IFN-α (20 ng/mL) for the indicated times before immunoblotting analysis was performed. Scale bars, 10 μm. Data are representative of three experiments with similar results

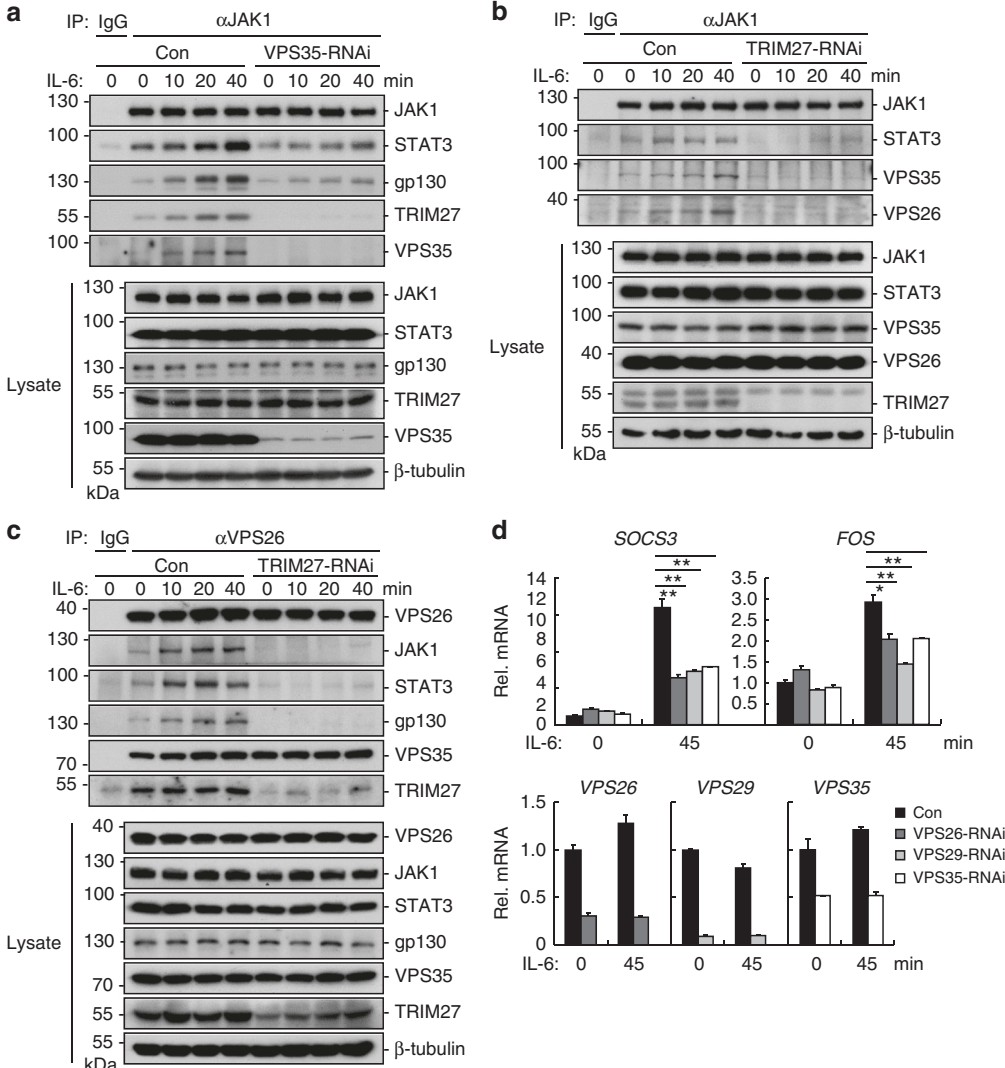

**Fig. 4** TRIM27 is required for recruitment of JAK1 and STAT3 to the retromer. **a** Effects of VPS35 knockdown on the association of JAK1 with STAT3, gp130, TRIM27, or VPS35. HeLa cells ($2 \times 10^7$) were starved overnight followed by stimulation with IL-6 (50 ng/mL) for the indicated times. Coimmunoprecipation and immunoblotting analysis were performed with the indicated antibodies. **b** Effects of TRIM27 knockdown on the association of JAK1 with VPS26, VPS35, or STAT3. HeLa cells ($2 \times 10^7$) were starved overnight followed by stimulation with IL-6 (50 ng/mL) for the indicated times. Coimmunoprecipation and immunoblot analysis were performed with the indicated antibodies. **c** Effects of TRIM27 knockdown on the association of VPS26 with JAK1, STAT3, gp130, VPS35, or TRIM27. HeLa cells ($2 \times 10^7$) were starved overnight followed by stimulation with IL-6 (50 ng/mL) for the indicated times. Coimmunoprecipation and immunoblotting analysis were performed with the indicated antibodies. **d** Effects of knockdown of the retromer components on IL-6-induced transcription of downstream genes. The control, VPS26-RNAi, VPS29-RNAi, and VPS35-RNAi stable HeLa cells ($2 \times 10^5$) were starved overnight and stimulated with IL-6 (20 ng/mL) for the indicated times before qPCR experiments. Data are representative of three experiments with similar results. Graphs show mean ± SD; $n = 3$. *$P < 0.05$, **$P < 0.01$, ***$P < 0.001$, unpaired $t$ test

deficient mice compared to their wild-type littermates (Fig. 8c). Immunoblot analysis further confirmed that the levels of phosphorylated STAT3 were markedly decreased in colon tumors in TRIM27-deficient mice, while there were no apparent differences in phosphorylated STAT3 levels in the tumor-adjacent normal colon tissues between wild-type and TRIM27-deficient mice (Fig. 8d). Consistently, the protein levels of STAT3 downstream target genes *Bcl-xl*, *c-Myc*, *Pcna*, and *Ccnd1*, which were responsible for tumor cell survival and proliferation, respectively[4,37], were dramatically downregulated in colon tumors of TRIM27-deficient mice compared to their wild-type littermates (Fig. 8d). Moreover, there was also a decrease in levels of inflammatory cytokines TNF-α, IL-1β, IL-6, and IL-17A in the sera or colon tissues of *Trim27*[−/−] mice (Fig. 8e), which was

consistent with the decreased inflammatory cell infiltrations in colon tumors of *Trim27*[−/−] mice (Fig. 8c). Collectively, these results suggest that TRIM27 plays important roles in CAC development.

**TRIM27 is overexpressed in human colorectal cancer (CRC) tissues**. To evaluate the clinical relevance of TRIM27 expression in human CRC, we analyzed the expression of TRIM27 in CRC tissues using the public data from the TCGA database (https://portal.gdc.cancer.gov/). We found that TRIM27 level was significantly higher in the CRC samples compared with normal colon tissues (Supplementary Figure 7a). In addition, the patients with high expression of TRIM27 had significantly

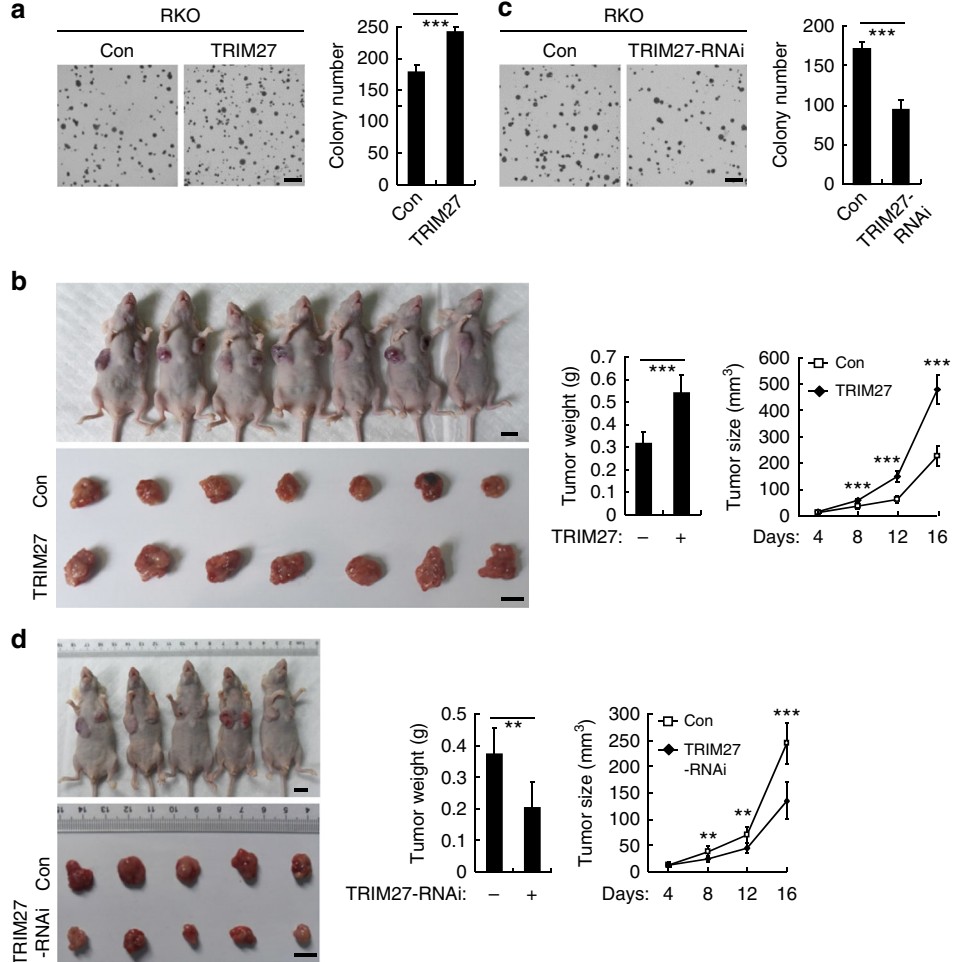

**Fig. 5** TRIM27 promotes growth and tumorigenicity of tumor cells. **a** Effects of TRIM27 on cell anchorage-independent growth. The control and TRIM27-overexpressing RKO cells ($2 \times 10^3$) were seeded in soft agar in 6-well plates. After 3 weeks, the colonies were photographed (left panel), and colony numbers were counted in each well (right panel). Results are represented as mean ± SD, $n = 3$. Scale bars, 3 mm. **b** TRIM27 promotes tumor growth in nude mice. The control and TRIM27-overexpressing RKO cells ($1 \times 10^7$) were injected into the flanks of nude mice. Mice were killed and photographed at 16 days after injection and tumor weights (upper panel) were measured. The tumor sizes (lower panel) were measured at an interval of 4 days after injection and calculated as $= \pi/6 \times L \times W \times H$, where $L$, $W$, and $H$ represent the length, width, and height of the tumor, respectively. Results are represented as mean ± SD, $n = 7$. Scale bars, 1 cm. **c** Effects of TRIM27 knockdown on cell anchorage-independent growth. The control and TRIM27-RNAi RKO cells ($2 \times 10^3$) were seeded in soft agar in 6-well plates. After 3 weeks, the colonies were photographed (left panel), and colony numbers were counted in each well (right panel). Results are represented as mean ± SD, $n = 3$. Scale bars, 3 mm. **d** Knockdown of TRIM27 inhibits tumor growth in nude mice. The control and TRIM27-RNAi RKO cells ($1 \times 10^7$) were injected into the flanks of nude mice. Mice were killed and photographed at 16 days after injection and the tumor weights (left panel) were measured. The tumor sizes (right panel) were measured at an interval of 4 days after injection and calculated as $V = \pi/6 \times L \times W \times H$, where $L$, $W$, and $H$ represent the length, width, and height of the tumor, respectively. Results are represented as mean ± SD, $n = 5$. Scale bars, 1 cm. *$P < 0.05$, **$P < 0.01$, ***$P < 0.001$, unpaired $t$ test. Data shown are from a representative experiment repeated for at least twice.

worse survival than those with low TRIM27 expression (Supplementary Figure 7b). These results suggest that increased TRIM27 expression is correlated with the pathogenesis of CRC in human.

## Discussion

STAT3 activity is tightly controlled for proper physiological processes, and dysregulation of STAT3 activity results in tumorigenesis. Aberrant and persistent activation of STAT3 can be mainly attributed to either oversupply of certain cytokines in tumor microenvironment or abnormal expression and dysfunctions of positive and negative regulators. For examples, elevated expression of STAT3 activators, such as IL-6, IL-11, and S1P, has been observed in various types of cancers[37,38]. In

addition, increased expression or aberrantly activated upstream tyrosine kinases, such as JAK1, JAK2, EGFR, and BMX, as well as other positive regulators, such as EZH2 and PASD1, also contribute to persistent activation of STAT3[5,32,39–41]. On the other hand, impaired function or reduced expression of negative regulators, such as PIAS3, PTPs, and GDX, also results in constitutive activation of STAT3[42,43]. In this study, we identified TRIM27 as an important mediator of STAT3 activation, which also plays a role in colitis and CAC development. More importantly, our study reveals that the retromer is an essential platform for STAT3 activation induced by cytokines such as IL-6 and OSM.

Overexpression of TRIM27 markedly potentiated IL-6- and OSM-induced STAT3 activation, whereas TRIM27 deficiency had opposite effects. Overexpression of TRIM27 marginally enhanced

IFN-γ-induced transcription of IRF1. It has been previously reported that TRIM27 negatively regulates IKKα/β-mediated NF-κB signals and IKKε-mediated IRF signals[31]. It is possible that the distinct cellular localization, post-translational modification, or complex formation of TRIM27 is responsible for its differential functions.

We found that overexpression of TRIM27 promoted growth and tumorigenicity of tumor cells in xenograft models. In addi-

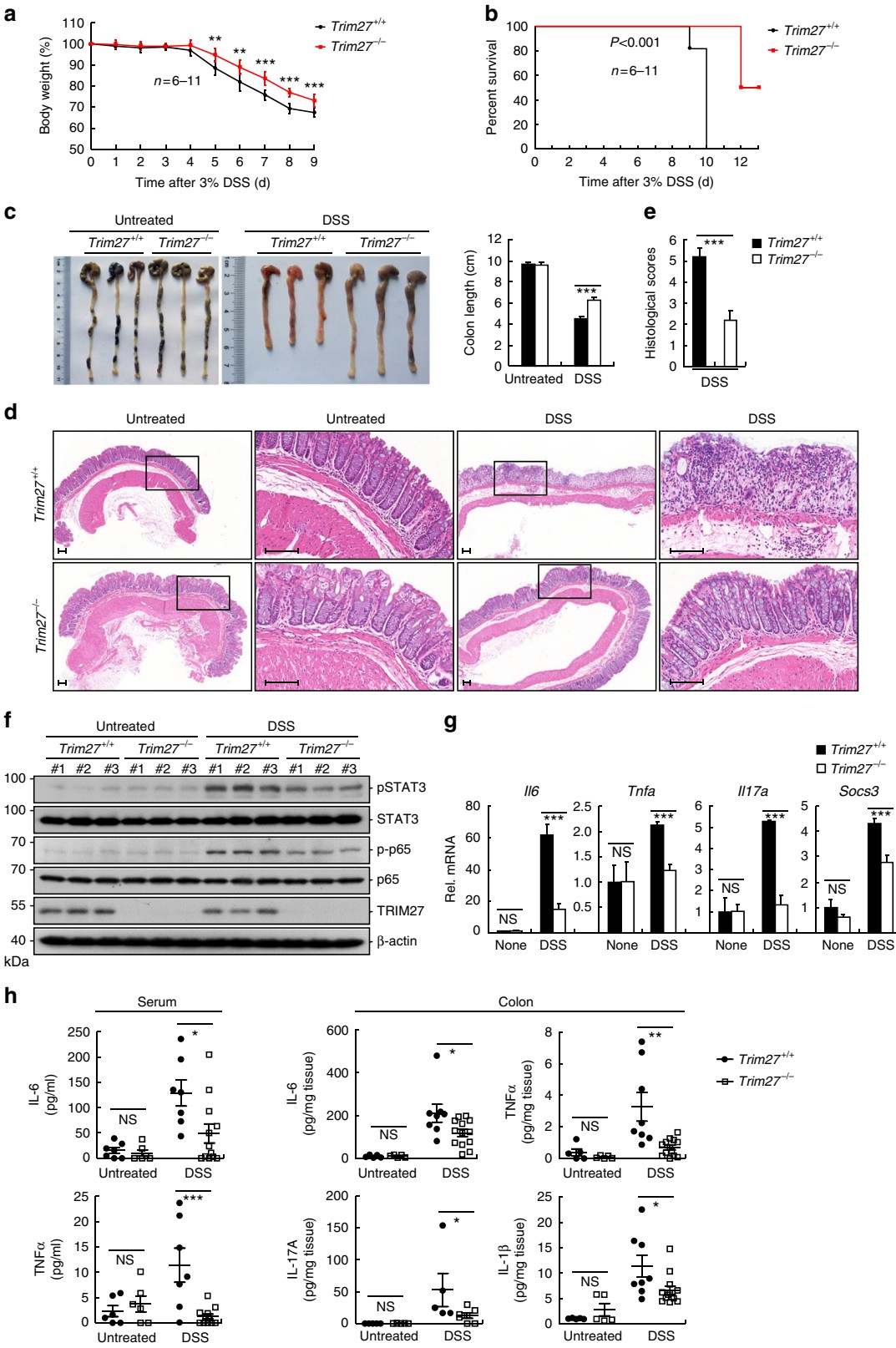

tion, TRIM27 levels are elevated in human CRC tissues, and the high expression levels correlate with adverse survival of CRC patients[44]. These results suggest that TRIM27 is a potential oncogenic protein, which is consistent with the observation that TRIM27 is highly expressed in other types of cancer, including seminomas, breast, endometrial, ovarian, lung, and colon cancers[19–23].

As a mediator of IL-6-induced STAT3 activation, TRIM27 plays important roles in DSS-induced colitis and AOM/DSS-induced CAC development. In the DSS-induced acute colitis model, $Trim27^{-/-}$ mice displayed attenuated colitis in comparison to their wild-type littermates. Consistently, TRIM27-deficient mice had decreased activation of STAT3 and levels of inflammatory cytokines including IL-6, TNFα, and IL-17A in the sera and/or colonic mucosa. In addition, our results suggest hematopoietic cells are responsible for attenuation of DSS-induced colitis in TRIM27-deficient mice. Previously, it has been observed that TRIM27 levels are increased in colon tissues of Crohn's disease patients[25] and in the $CD4^+$ T lymphocytes of mesenteric lymph nodes in DSS-induced colitis model[45]. These studies collectively point to an important role of TRIM27 in colitis.

Chronic or unresolved inflammation promotes tumorigenesis. In the AOM/DSS model, TRIM27-deficient mice developed less, smaller, and lower-grade colon tumors compared with their wild-type littermates. Consistently, the tumor tissues in TRIM27-deficient mice had decreased STAT3 activation, lower levels of inflammatory cytokines, lower infiltration of inflammatory cells, and decreased induction of STAT3 target genes including $Bcl-xl$, $c-Myc$, $Pcna$, and $Ccnd1$. These results suggest that TRIM27 plays important roles in CAC development.

In addition, we found that the body weight changes and inflammatory cytokine production between WT and TRIM27-deficient mice during AOM/DSS treatment were not as notable as that of during the process of DSS-induced acute colitis, indicating that loss of TRIM27 barely reduced colonic inflammation in AOM/DSS-treated mice. Moreover, it has been reported that TRIM27 acts as a negative regulator of type I IFNs production in response to virus infection[31,46]. However, there is no difference in type I IFN production between TRIM27-deficient mice and their wild-type littermates, indicating that the actual link between TRIM27 and STAT3 seems to be direct. As TRIM27 could function both in hematopoietic cells and IECs to regulate STAT3 activation, less mucosal inflammation orchestrated by hematopoietic cells and impaired activation of STAT3 as well as expression of pro-oncogenic STAT3 target genes in IECs may lead to development of fewer tumors in TRIM27-deficient mice in this model.

Our findings reveal unexpected functions of the retromer as an important platform for IL-6-induced STAT3 activation. Confocal microscopy indicated that TRIM27 was localized at the retromer-positive structures but not early endosomes. Interestingly, phosphorylated STAT3 induced by IL-6 stimulation was also colocalized with TRIM27 at the retromer-positive structures but not early endosomes. Knockdown of TRIM27 impaired the recruitment of STAT3 to JAK1 as well as gp130, JAK1, and STAT3 to the retromer but not the assembly of the retromer, suggesting that TRIM27 acts as a linker for recruiting gp130, JAK1, and STAT3 to the retromer. Consistently, knockdown of the retromer component VPS35 impaired the associations of TRIM27 with JAK1 and STAT3, and inhibited IL-6-induced transcription of downstream genes. These results suggest that the retromer-positive structure acts as a platform for TRIM27-mediated assembly of JAK1–STAT3 complexes and subsequent phosphorylation of STAT3 following IL-6 stimulation.

Previously, it has been reported that internalized IFNAR1 and IFNAR2 subunits of the IFNAR complex are differentially sorted by the retromer at the early endosome, in which IFNAR2 is recycled to the plasma membrane, whereas IFNAR1 is sorted to the lysosome for degradation[36]. Knockdown of VPS35 leads to abnormally prolonged residency and association of the IFNAR subunits at the early endosome, resulting in prolonged activation of STAT1 in response to type I IFNs[36]. In light of this observation, it seems that the retromer can act as a platform for activation of different members of the STAT family, though by different mechanisms. It has also been reported that TRIM27-MAGEL2 is required for retrograde transport via facilitating K63-linked ubiquitination of WASH, which is an essential regulator of retromer-mediated transport[33]. However, our results indicated that the E3 ligase activity of TRIM27 was dispensable for mediating STAT3 activation, suggesting that different signals can dictate how TRIM27 functions in the retromer.

It has been reported that c-Met can activate STAT3 at the endosome and phosphorylated STAT3 induced by c-Met is colocalized with EEA1, an early endosome marker. Our results indicated that phosphorylated STAT3 induced by IL-6 was localized at the retromer-positive structures but not the endosome, which is consistent with a previous study showing that phosphorylated STAT3 induced by IL-6 does not colocalize with various endosomal markers including EEA1, Rab5, Rab7, Rab11, and LAMP1[15]. These studies suggest that different signaling pathways may utilize different platforms for activation of STAT3. Since TRIM27 deficiency also inhibited STAT3 activation by OSM, which is another IL-6 family member, it is possible that the retromer is a common platform for STAT3 activation induced by the IL-6 family members which share the common co-receptor gp130. Our findings provide important insights to the mechanisms of STAT3 activation induced by the IL-6 family and identify TRIM27 as a potential therapeutic target for treatment of cancer.

**Fig. 6** Deficiency of TRIM27 attenuates DSS-induced colitis. **a** $Trim27^{+/+}$ ($n = 11$) and $Trim27^{-/-}$ ($n = 6$) mice were treated with 3% DSS over a 10-day period, and their body weights were daily monitored. Results are represented as mean ± SD. **b** Survival of mice described in **a** was monitored until the 13th day. Results are represented as mean ± SD. **c** $Trim27^{+/+}$ and $Trim27^{-/-}$ mice were treated with 3% DSS or left untreated for 9 days before their colon lengths were measured. Results are represented as mean ± SD, $n = 3$. **d** Representative images of haematoxylin and eosin staining of colon tissues of mice described in **c**. Scale bars, 100 μm. **e** Histological analysis of colon tissues described in **d**. The histological scores were determined in a double-blind manner. Results are represented as mean ± SD, $n = 5$. **f** Immunoblotting analysis of pY705-STAT3 and p-P65 levels in colon tissues of $Trim27^{+/+}$ and $Trim27^{-/-}$ mice treated with 3% DSS or left untreated for 9 days. Lysates from three different mice were analyzed for each group. **g** qPCR analysis of colon tissues of $Trim27^{+/+}$ and $Trim27^{-/-}$ mice treated with 3% DSS or left untreated for 9 days. Results are represented as mean ± SD, $n = 3$. **h** ELISA measurement of cytokine levels in sera and colon tissues of $Trim27^{+/+}$ and $Trim27^{-/-}$ mice treated with 3% DSS or left untreated for 9 days. Results are shown as mean ± SD, $n = 5$–13. *$P < 0.05$, **$P < 0.01$, ***$P < 0.001$, unpaired $t$ test (**a**, **c**, **e**, **g**, **h**) or log-rank test (**b**). Data are representative of three experiments with similar results

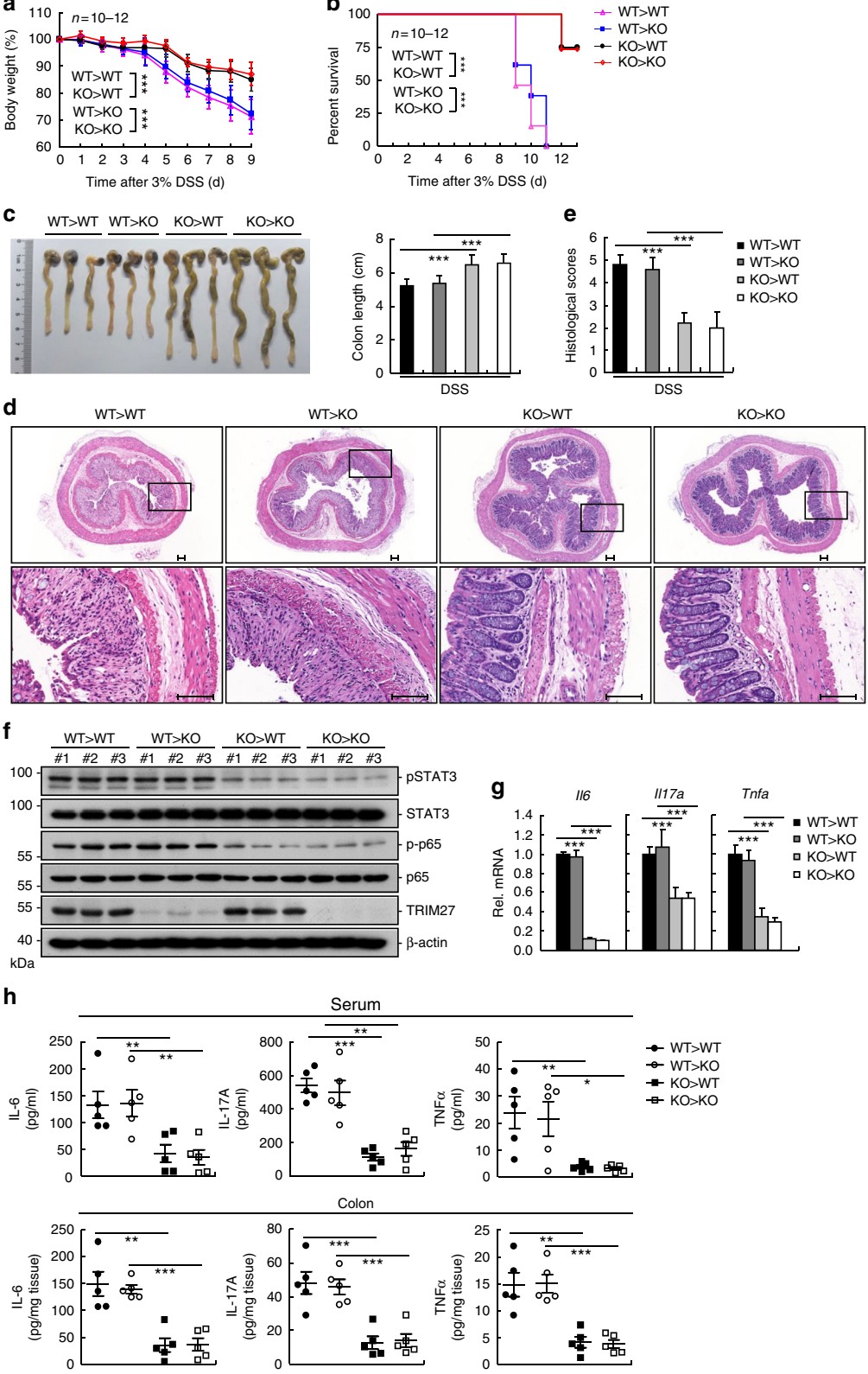

## Methods

**Mice.** $Trim27^{-/-}$ mice on the C57BL/6 background were generated by the CRISPR/Cas9 method[47]. $Trim27^{-/-}$ mice bearing a 26 bp deletion in its exon-2 were identified using primers flanking the break sites. The sequences of the primers are 5′-CTTCGTGGAGCCTATGATGC-3′ (forward), and 5′-TCGCAGTACAG CTTCAGAGG-3′ (reverse). All mice were housed under specific pathogen-free conditions at the animal facilities of Wuhan University College of Life Sciences and Wuhan Institute of Virology of the Chinese Academy of Sciences. Eight–ten-week old, age- and sex-matched mice were used in all the experiments. All mouse studies were approved by the Animal Care Committees of Wuhan University College of Life Sciences and Wuhan Institute of Virology of the Chinese Academy of Sciences.

**Reagents, antibodies, and cells.** Recombinant human IL-6, OSM, INF-β, IFN-γ, and mouse IL-6 (Peprotech); mouse monoclonal antibodies against Flag (1:1000 for immunoblots (IB), 1:200 for immunofluorescent staining (IF)) and β-actin (1:1000 for IB) (Sigma), β-tubulin (Invitrogen, 1:1000 for IB), HA (Covance, 1:1000 for IB), PCNA (F-2, sc-25280, 1:1000 for IB) and VPS35 (B-5, sc-374372, 1:1000 for IB) (Santa Cruz Biotechnology); rabbit monoclonal antibodies against pY705-STAT3 (D3A7, 1:1000 for IB, 1:100 for IF), BCL-XL (#2764, 1:1000 for IB), c-MYC (D84C12, 1:1000 for IB), and p-P65 (S536, 1:1000 for IB) (Cell Signaling Technology); rabbit polyclonal antibodies against STAT3 (C-20, sc-482, 1:1000 for IB, 1:200 for IF) and p65 (C-20, sc-372, 1:1000 for IB) (Santa Cruz Biotechnology), JAK1 (#610232, BD, 1:500 for IB), VPS26 (ab-23892, Abcam, 1:1000 for IB), and TRIM27 (#18791, IBL, 1:500 for IB) were purchased from the indicated companies. Mouse and rabbit polyclonal antibodies against gp130 were raised against N-terminal portion of human gp130 protein.

HEK293, HeLa, HT29, and RKO cells were obtained from ATCC. For all the cells used, mycoplasma contamination was checked.

**Constructs.** Mammalian expression plasmids for Flag- or HA-tagged TRIM27 and its mutants, gp130, JAK1, and its mutants, JAK2 and its mutants, STAT3 and its mutants, VPS35, VPS29, VPS26, and GFP- or Cherry-tagged TRIM27, VPS35, VPS29, and VPS26 were constructed by standard molecular biology techniques. STAT3 and STAT1/2 luciferase reporter plasmids were purchased from Qiagen. IRF1 luciferase reporter plasmid was previously described[48].

**Transfection and luciferase reporter assays.** HEK293 cells were transfected by standard calcium phosphate precipitation method. HeLa and RKO cells were transfected by Fugene HD (Roche). Empty control vector was added to ensure that each transfection receives the same amount of total DNA. For normalization of transfection efficiency, 0.01 μg of pRL-TK (Renilla luciferase) reporter plasmid was added to each transfection. Luciferase assays were performed using a dual-specific luciferase assay kit (Promega). All reporter assays were repeated at least three times.

**RNAi experiments.** Double-stranded oligonucleotides corresponding to the target sequences were cloned into the pSuper.Retro RNAi plasmid (Oligoengine). The following sequences were targeted for human TRIM27 mRNA: #1—5′-GCTGAACTCTTGAGCCTAA-3′; #2—5′-CGGAGAGTCTAAAGCA GTT-3′; #3—5′-GACTCAGTGTGCAGAAAAG-3′. A pSuper.retro RNAi plasmid targeting green fluorescent protein (GFP) mRNA was used as a control for all RNAi-related experiments.

**qPCR.** Total RNA was isolated from cells using TRIzol reagent (TAKARA). After reverse transcription, the cDNA products were subjected to real-time PCR analysis to measure mRNA expression levels of tested genes. Gene-specific primer sequences were listed in Supplementary Table 1.

**Establishment of stable cell lines.** HEK293 cells were transfected with TRIM27-overexpressed or TRIM27-RNAi or corresponding control retroviral plasmids and two packaging plasmids (pGag-Pol and pVSV-G) by calcium phosphate precipitation. The culture medium was replaced with new medium without antibiotics at 12 h after transfection. After additional 24 h, the recombinant virus-containing medium was filtered and used to infect HeLa, RKO, or HEK293 cells in the presence of polybrene (8 μg/mL). The infected cells were selected using puromycin (2 μg/mL) for 2 weeks before additional experiments were performed.

**Coimmunoprecipitation and immunoblot analysis.** For transient transfection and coimmunoprecipitation experiments, HEK293 cells ($1 \times 10^6$) were transfected for 18–24 h. The transfected cells were lysed in 1 mL of lysis buffer (20 mM Tris, pH 7.5, 150 mM NaCl, 1% Triton X-100, 1 mM EDTA, 10 μg/mL aprotinin, 10 μg/mL leupeptin, 1 mM phenylmethylsulfonyl fluoride, 1 mM NaVO₃, 2.5 mM sodium pyrophosphate, and 1 mM β-glycerol-phosphate). For each immunoprecipitation, a 0.4-mL aliquot of the lysate was incubated with 0.5 μg of the indicated antibody or control IgG and 25 μL of a 1:1 slurry of Protein G Sepharose (GE Healthcare) for 2 h. Sepharose beads were washed three times with 1 mL of lysis buffer containing 0.5 M NaCl. The precipitates were analyzed by standard immunoblot procedures.

For endogenous coimmunoprecipitation experiments, HeLa ($5 \times 10^7$) cells were starved in DMEM without FBS overnight and stimulated with IL-6 (100 ng/mL) for the indicated times or left untreated. Coimmunoprecipitation and immunoblot experiments were performed as described above. Uncropped scans of the immunoblots are provided in Supplementary Figures 8–14.

**Confocal microscopy.** HeLa cells were transfected with the indicated plasmids by Fugene HD (Roche). At 24 h post transfecton, the cells were starved with medium without FBS overnight and then treated with IL-6 (50 ng/mL) for the indicated times followed by fixation with 4% (w/v) paraformaldehyde for 15 min at 4 °C. The fixed cells were then permeabilized in PBS containing 0.1% Triton X-100 for 5 min. After washing in PBS, cells were blocked with 1% BSA prepared in PBS for 1 h, and then incubated with the indicated antibodies overnight at 4 °C. After washing with PBS, cells were incubated with the secondary antibodies conjugated with Alexa Fluor 594 or Alexa Fluor 647 dye for 1 h at room temperature before confocal microscopy.

**Soft agar assay.** TRIM27-overexpressed or TRIM27-RNAi or corresponding control HeLa/RKO/HEK293 cells were suspended in 2 mL of 0.35% agarose in DMEM and seeded in each well of 6-well plate containing 2 mL of 0.7% agarose in DMEM. Cells were grown at 37 °C for about 2–3 weeks and stained with crystal violet (0.005% w/v).

**Tumor xenografts.** TRIM27-overexpressing or TRIM27-RNAi or corresponding control HeLa cells ($1 \times 10^6$) or RKO cells ($1 \times 10^7$) were injected into 6-week-old athymic BALB/c nu/nu mice. Tumor volume (V) was measured every 4 days. All mice were handled in accordance with the Wuhan University Medical Research Institute animal care and use committee guidelines.

**Colitis and CAC models.** For colitis model, sex- and age-matched $Trim27^{+/+}$ and $Trim27^{-/-}$ mice (8–10 weeks of age) were provided 3% DSS in their drinking water for 10 days. For CAC model, mice were injected intraperitoneally with AOM (10 mg/kg) and after 5 days, DSS (2.5%) was given in drinking water for 7 days followed by regular water for 14 days. This cycle was repeated twice with 2.5% DSS, and mice were killed on day 80. Histological assessments of colitis and severity scores were made in a double-blinded manner after hematoxylin and eosin (H&E) staining as described[49].

**Bone marrow chimeras.** To evaluate the contribution of hematopoietic cells to DSS-induced colitis, bone marrow chimeric mice were established as described[50]. In brief, 8-week-old recipient mice were lethally irradiated (single dose of γ irradiation, 950 rad) and intravenously injected with $10^7$ bone marrow cells isolated from wild-type and TRIM27-deficient mice 24 h after irradiation. The bone marrow reconstitution was performed reciprocally, resulting in the generation of four

**Fig. 7** Deficiency of TRIM27 in hematopoietic cells is responsible for attenuation of DSS-induced colitis. **a** Four groups of mice (WT > WT, WT > KO, KO > WT, and KO > KO) generated by bone marrow transplantations were treated with 3% DSS over a 9-day period, and their body weights were daily monitored. Results are represented as mean ± SD, $n = 10$–12. **b** Survival of mice described in **a** was monitored until the 13th day. Results are represented as mean ± SD, $n = 10$–12. **c** Colon lengths of mice described in **a** were measured 9 days after 3% DSS treatment. Results are represented as mean ± SD, $n = 7$–12. **d** Representative images of haematoxylin and eosin staining of colon tissues of mice described in **c**. Scale bars, 100 μm. **e** Histological analysis of colon tissues described in **d**. The histological scores were determined in a double-blind manner. Results are represented as mean ± SD, $n = 5$. **f** Immunoblotting analysis of pY705-STAT3 and p-P65 levels in colon tissues of mice described in **c**, $n = 3$. **g** qPCR analysis of colon tissues of mice described in **c**. Results are represented as mean ± SD, $n = 3$. **h** ELISA measurement of cytokine levels in sera and colon tissues of mice described in **a**. Results are represented as mean ± SD, $n = 5$. *$P < 0.05$, **$P < 0.01$, ***$P < 0.001$, unpaired $t$ test (**a**, **c**, **e**, **g**, **h**) or log-rank test (**b**). Data are representative of two experiments with similar results

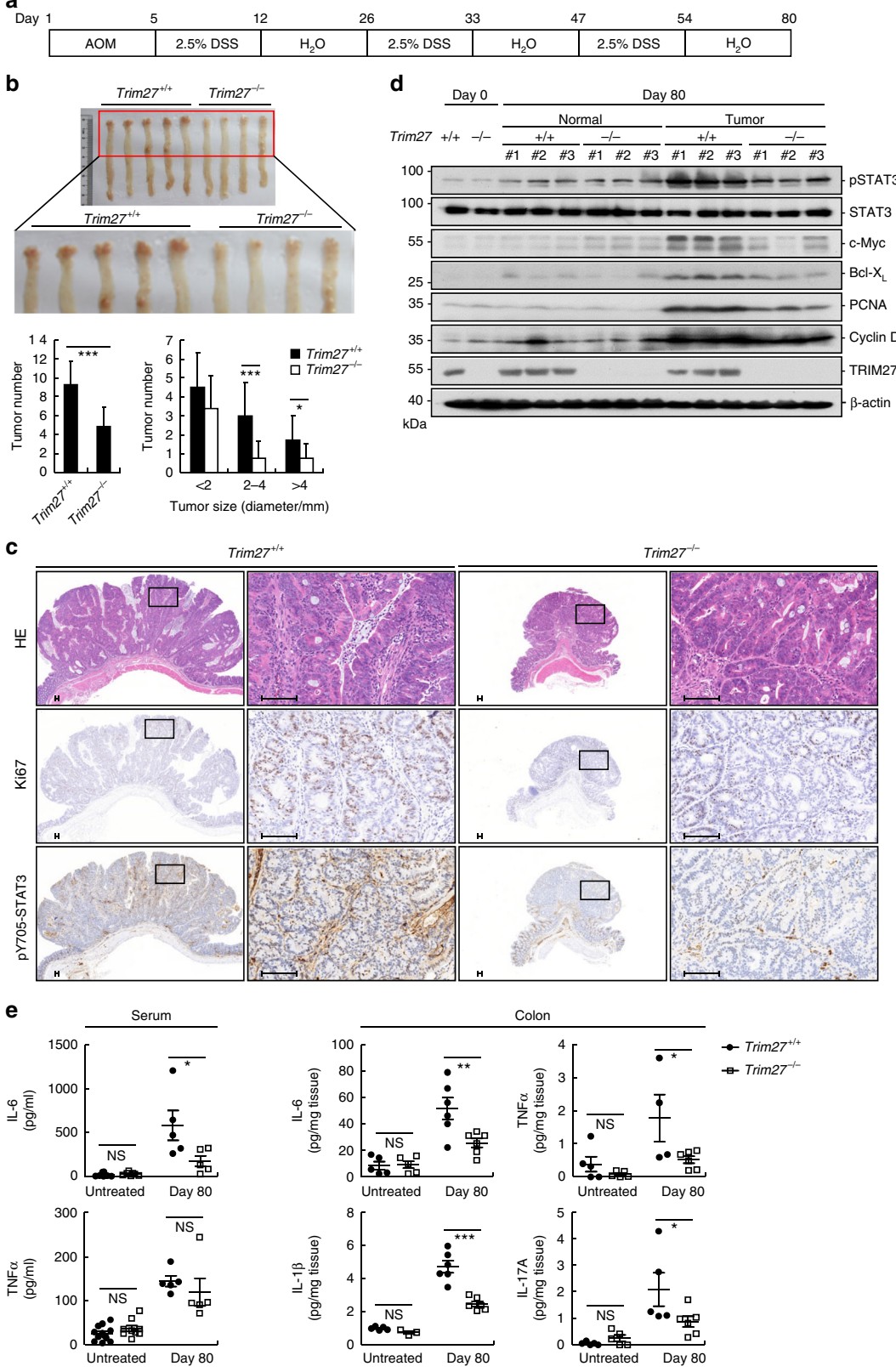

**Fig. 8** TRIM27 deficiency suppresses AOM/DSS-induced CAC. **a** A schematic overview of the CAC model. Five days after initial AOM injection (10 mg/kg), DSS (2.5%) was given in drinking water for 7 days followed by regular water for 14 days, and this cycle was repeated twice before the following analysis. **b** The colons of $Trim27^{+/+}$ and $Trim27^{-/-}$ mice were removed and photographed. The tumor numbers (left panel) and tumor sizes (right panel) were measured. Results are represented as mean ± SD, $n = 16$. **c** Representative images of immunohistochemical staining of colon tumors of $Trim27^{+/+}$ and $Trim27^{-/-}$ mice. Scale bars, 100 μm. **d** Immunoblotting analysis of colon tumors and adjacent normal tissues of $Trim27^{+/+}$ and $Trim27^{-/-}$ mice. Samples from three independent mice for each group were analyzed. **e** ELISA measurement of cytokine levels in sera and colon tissues of $Trim27^{+/+}$ and $Trim27^{-/-}$ mice. Results are represented as mean ± SD, $n = 5$–11. $*P < 0.05$, $**P < 0.01$, $***P < 0.001$, unpaired $t$ test. Data are representative of three experiments with similar results

---

groups of experimental mice. Recipient mice were housed for 8 weeks to fully reconstitute bone marrow before being treated with DSS.

**Statistics**. Unpaired Student's $t$ test was used for statistical analysis with Microsoft Excel and GraphPad Prism Software. For the mouse survival study, Kaplan–Meier survival curves were generated and analyzed by log-rank test; $P < 0.05$ was considered significant.

**Data availability**. The data that support the findings of this study are available within the article and its Supplementary Information or from the corresponding authors on reasonable request.

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

## Acknowledgements

We thank Dr. Hongliang Li (Wuhan University) for generation of *Trim27* knockout mice. This work was supported by the National Key R&D Program of China (2017YFA0505800 and 2016YFA0502102), the National Natural Science Foundation of China (91429304, 31630045, 31671465, 31521091, and 31700758), by the National Science Fund for Distinguished Young Scholars (31425010), and the Key Research Programs of Frontier Sciences funded by the Chinese Academy of Sciences.

## Author contributions

H.-B.S., Y.-Y.W., H.-X.Z. and Z.-S.X. conceived and designed the study; H.-X.Z., Z.-S.X., H.L., M.L., T.X., K.C. and S.-Y.W. performed the experiments; H.-B.S., Y.-Y.W., H.-X.Z., Z.-S.X. and Y.L. analyzed the data; H.-B.S., Y.-Y.W., H.-X.Z. and Z.-S.X. wrote the manuscript.

## Additional information

**Competing interests:** The authors declare no competing interests.

