## [Peer Review File · Nature Communications]

Reviewers' comments:

Reviewer #1 (Remarks to the Author):

TRIM proteins have a number of biological functions including immunological signals, cell proliferation and differentiation, gene expression. Recent accumulating evidence indicates several TRIM proteins serve as important regulators of intracellular signals in immunity and carcinogenesis. TRIM27, one of the TRIM family proteins, has also been reported as one of regulators for spermatogenesis, IFN signals, innate immunity, inflammation, neuronal protection and carcinogenesis.

In this paper, the authors showed that TRIM27 regulates the STAT3-mediated signal pathway. The authors found that TRIM27 interacts with STAT3 and retromer-related proteins; thereby leading to enhancement of STAT3-mediated signals and that TRIM27 exhibits the inflammation and carcinogenesis in colon via DSS administration. These findings suggest that TRIM27 is a critical positive regulator for the etiology of inflammation- mediated carcinogenesis in colon.

These insights are relevant in understanding the novel roles of TRIM27 in regulation of the STAT3-mediated signal pathway. This manuscript contains some important issues, such as the molecular insight of TRIM27 in the interaction with retromers. Although almost experiments have been well performed and described, there may be some overstatements of results that preclude publication of the manuscript in the present form.

Specific points

1. The most important issue is to clarify the molecular mechanism by which TRIM27 regulates the activation pathway including STAT3 (including STAT3-related molecules such as Jaks) and retromer-related proteins.
2. Fig. 1B: It has been previously reported that TRIM27 negatively regulates IKK α /b-mediated NF κ B signals and IKK ϵ -mediated IRF signals. However, the authors showed that TRIM27 activated IL-6-mediated signals. Did the authors check several signal pathways other than the signals through IL-6 or IFN- β ? The authors should mention the molecular insights for the difference among these signals in the discussion section.
3. Fig. 1G: anti-TRIM27 immunoblots is poor or unreliable. The authors should clearly show anti-TRIM27 immunoblots. The bands were scraped in this manuscript.

Reviewer #2 (Remarks to the Author):

This manuscript addresses the mechanisms controlling STAT3 activation, a TF phosphorylated by non-receptor Janus tyrosine kinase (JAK) in response to diverse ligands, such as GF or interleukins (IL); given the downstream genes becoming upregulated, this signaling pathway is of relevance to cancer and inflammation.

The major finding is the implication of the tripartite motif (TRIM) family member TRIM27 in a protein complex with JAK1 and activated STAT3 induced by IL6 family members. Importantly, the authors show evidence that the endosomal sorting complex retromer is required for the JAK1/TRIM27/STAT3 association and consequent transcription activation of downstream genes induced by IL6.

Data are obtained using ingenious strategies and supported by various model systems. In general, evidence provided is strong and supports the conclusions. Overall, the manuscript is clearly

written.

Points that should be addressed are the following:

- In the title, and in many instances throughout the manuscript, it is stated that "TRIM27 mediates STAT3 activation at the retromer..." The authors should specify it either as 'retromer-positive structures' or 'compartment containing retromer'. Thus, stating in line 337 that "TRIM27 was localized at the retromer but not endosomes" is not accurate, since they meant localization to a retromer-positive compartment as opposite to early endosomes.

Along these lines, when retromer is defined on line 107 as "a highly conserved heteropentameric complex important for recycling proteins from the endosomes..." nowhere in the manuscript the complex is properly defined. This would be appropriate in the introduction as well as in the discussion. Indeed, the Vacuolar protein sorting (Vps) complex Vps26/Vps35/Vps29 is a trimer considered – at least till recently – as the main cargo-selective subcomplex of retromer, yet the referred heteropentamer includes an interacting sorting nexin dimer, whose potential role is not addressed here.

- Lines 222 & 345: "retromer acts as a platform for TRIM27-mediated assembly of JAK1-STAT3 complexes and subsequent activation of STAT3 following IL-6 stimulation." The proposed role of retromer "as a platform" is recurrent throughout the text. Could perhaps the evidence provided indicate that retromer functions here in recruitment for sorting of signaling molecules or simple defines a location where the signaling molecules act? In other words, is it actively participating in TRIM27-mediated delivery of pSTAT3 toward the nucleus?

- The immunofluorescence data are somewhat poor, in part because images shown are at insufficient magnification; perhaps an overlap of endogenous Vps26 with TRIM27 and especially with pSTAT3 seen at higher magnification would help; there should be commercially available antibodies against Vps26 suitable to use on cells.

- Where cells in Fig 1A, B & C treated with IL6 for 10 h? This sounds in striking contrast with the 0-60 min treatment indicated for instance in Fig 1D.

- Please ensure to indicate the cell model used in all figures and/or the corresponding legends. For instance, are data in Fig 1E & G obtained in HeLa cells as Fig 1F? Fig S3 legend appears also misleading.

- Blots display changes that are often not striking, and quantitation is unnecessary. Yet, please state in legends that a representative experiment is shown.

- Line 81, please rewrite "into lipid vesicles called endosomes"; why not simply 'cellular vesicles'?

- A few typo errors should be fixed; e.g. please replace in line 103 "reported to involve in" for "involved in". Verb tenses should also be double checked.

- Line 198, and in other locations: please replace "punctuate" for "punctate".

- Line 198: please take off "or microsomes", since this term refers to ER-derived artefactual structures obtained by cell fractionation.

- Line 207: "knockdown of VPS35 prolonged IFN-alpha-induced STAT1 phosphorylation (Fig 3E), which is consistent with a previous report (Ref 34)." Yes, but keep in mind that in Ref 34-Fig 7A & B the effect (increase of approx. 50 %) is seen only at 60 min from induction, not before that.

- Line 217: The sentence: "These data suggest that the..." may be removed due to redundancy

with that at the end of the paragraph.

- Fig 5: Why is there such a high discrepancy in controls if the same experimental conditions were used? Compare A with C for HeLa cells (colony # goes from 50 to almost 200), and B with D for the mouse model (similarly, for tumor growth).

- Fig. 6E & 7D: are 1, 2, 3 triplicates?

- Fig 7D: a blot for Cyclin D1 is shown, however not mentioned in the text.

- Line 339: "...recruitment of STAT3 to JAK1 as well as gp130, JAK1 and STAT3 to the retromer but not the assembly of the retromer, suggesting that TRIM27 acts as a linker for recruiting gp130, JAK1 and STAT3 to the retromer." (See also line 41 in the Abstract.) This sentence is confusing and should be rephrased. In particular, the interaction with gp130 is not shown in Fig 4. According to the data presented, retromer does not interact with gp130 (Fig 3D), whereas TRIM27 does it (Fig 2D).

- Line 470: the formula used to calculate tumor volume appears convenient, yet may not be accurate since most tumors have hemisphere (or ellipsoid) shape; $W = H$ according to the formula used. Why was not used instead $V = (\pi \text{ number}/6) \times L \times W \times H$ (or expressed differently, $V = (4/3) \pi \times r_1 \times r_2 \times r_3$).

- Line 553: please add pg. numbers in Ref. 25.

- Line 898: please replace "as schematic shown" for "as in the schematic diagram shown" or "as in the schema shown".

- Line 906: "minutes" should refer to "min".

Reviewer #3 (Remarks to the Author):

In this work, Zhang et al identify TRIM27 as a positive regulator of IL-6-induced STAT3 activation. Their data show that TRIM27 is required for JAK1-STAT3 complex formation. In addition, their data show that deletion of TRIM27 reduces DSS-induced acute inflammation and AOM/DSS-induced tumorigenesis. Although this study is interesting, some of key conclusions are not fully supported by the present experiments as outlined in more detail below.

Major points:

1. Most in vitro experiments were done in one cell line, HEK293. The key experiments should be performed in more than one cell line. Since the authors showed the function of TRIM27 in acute colonic inflammation and colitis-associated tumorigenesis, they should confirm their in vitro results in colon epithelial cells and/or colonic stroma cells.

2. Although the authors present data showing that IL-6 fails to induce activation of STAT3 and transcription of its downstream genes in TRIM27-deficient BMDMs (Fig. 1H-I) and in TRIM27-deficient primary hepatocytes, it is not clear whether IL-6 also fails to induce activation of STAT3 and transcription of its downstream genes in colonic epithelial cells and/or stroma cells.

3. DSS-treated and DSS/AOM-treated mice are not good models for IBD and IBD-associated CRC because the DSS model represents a process of injury and wound healing. Therefore, the role of TRIM27 in IBD and IBD-associated CRC should be evaluated in other mouse models of IBD and IBD-associated CRC.

4. It seems that the results from Fig. 2A-B indicate that TRIM27 can replace TYK2 in IL-6-induced STAT3 activation. Is this correct? Does TRIM27 interact with TYK2?

5. In Fig. 2D, does IL-6 really induce the association of TRIM27 with gp130 and JAK2. Similarly, it's hard to see how IL-6-induced the association of JAK1 with TRIM27 and VPS35 in Fig. 4A. much better images should be provided.

6. Does knockdown of TRIM27 reduce the association of VPS35 with JAK1 and STAT3?

7. Since the error bars have wide variations in Fig. 7B, it's necessary to confirm that the difference of tumor numbers between WT and TRIM27-deficient mice is significant.

8. Does loss of TRIM27 reduce colonic inflammation in AOM/DSS-treated mice?

Minor points:

1. Does knockdown of TRIM27 also inhibit IL-6-induced STAT3 transactivation (STAT3 report activity)?

2. Does overexpression of TRIM27 in HEK293 promote cell anchorage-independent growth and tumor growth in nude mice?

Reviewer #4 (Remarks to the Author):

In this manuscript the authors investigated the possible interaction between TRIM27 and STAT3 signaling. Overexpression of TRIM27 led to upregulation of phosphorylated STAT3 activation. Mice deficient in TRIM27 appeared to be less susceptible to acute DSS colitis. STAT3 activation in whole colonic tissues of colitic TRIM27^{-/-} seemed to be less when compared to TRIM27^{+/+}. In addition, TRIM27^{-/-} developed fewer tumors in the AOM/DSS model and these smaller tumors showed less STAT3 phosphorylation.

While the observations are novel and interesting, three key questions must be addressed:

1) What cell type is crucial for STAT3 activation via TRIM27 in acute DSS colitis? STAT3 plays cell type-dependent roles in the inflamed gastrointestinal tract. Is TRIM27-induced STAT3 activation myeloid-specific or IEC-specific? STAT3 in IEC maintains barrier integrity and survival, while STAT3 in myeloid cells may mediate anti-inflammatory effects and promote formation of colitis-associated tumors.

2) TRIM27 has previously been shown to act as a negative regulator of type I IFN production. Signaling through type I IFN receptors may regulate the activation of STAT3. In animal models of IBD it has been shown that type I IFN may protect against colitis. Is it possible that the actual link between TRIM27 and STAT3 is INDIRECTLY through altered type I IFN production - at least in vivo?

3) Did TRIM27^{-/-} develop fewer tumors because they just developed less mucosal inflammation in this model?

Minor:

Histology score is missing in Figure 6.

In each figure/panel, it must be clearly indicated how many mice per group were investigated per experiment/assay.

Please indicate in MM whether mice were age/gender-matched and whether mice were specific pathogen-free. Viral co-triggers have been shown to influence acute DSS colitis in animal facilities. TRIM27 may play a role in virus-host interactions - therefore please indicate in MM that mice were specifically virus-free.

Following is a point-to-point response to the reviewers' concerns.

Reviewer #1:

Major comments

.....

These insights are relevant in understanding the novel roles of TRIM27 in regulation of the STAT3-mediated signal pathway. This manuscript contains some important issues, such as the molecular insight of TRIM27 in the interaction with retromers. Although almost experiments have been well performed and described, there may be some overstatements of results that preclude publication of the manuscript in the present form.

Specific points

1. The most important issue is to clarify the molecular mechanism by which TRIM27 regulates the activation pathway including STAT3 (including STAT3-related molecules such as Jaks) and retromer-related proteins.

Reply:

In addition to the experiments shown in the original manuscript, we have now performed additional experiments to clarify the molecular mechanisms by which TRIM27 regulates the activation pathway including STAT3 and retromer-related proteins..

Firstly, to examine whether endocytosis is required for IL-6-induced STAT3 activation, we pretreated HEK293 cells with dynasore (an inhibitor of dynamin)¹ for 30 min before IL-6 stimulation, and then performed reporter assays and immunoblotting analysis. As shown in the new Fig. S4E&F and described in page 10, IL-6-induced activation of STAT3 as well as phosphorylation of STAT3 at Y705 was markedly inhibited by dynasore. These results indicate clathrin-dependent endocytosis of IL-6R α /gp130 receptor complex is required for IL-6-induced STAT3 activation.

Secondly, our original results showed that retromer component VPS35 could interact with JAK1, JAK2 and STAT3 but not IL6-R α or gp130 in overexpression systems (Figure 3D). We have now performed additional coimmunoprecipitation experiments to detect whether VPS35 interacts with endogenous signaling components following IL-6 stimulation. As the antibody against VPS35 could not enrich endogenous VPS35, we generated a HeLa cell line in which relative low level of Flag-tagged VPS35 was expressed. We treated this cell line with IL-6 for different time and performed coimmunoprecipitation experiments using anti-Flag antibody. As shown in the new Figure 3E and described in page 10, the association of VPS35 with the signaling components gp130, JAK1 and STAT3 was induced by IL-6 stimulation, whereas VPS35 constitutively interacted with VPS26 and TRIM27 (new Figure 3E). In addition, the previously prepared samples immunoprecipitated by anti-VPS26 were subjected to immunoblotting analysis with anti-gp130. As shown in new Fig. 4C and described in page 11, the association of VPS26 with gp130 was enhanced following IL-6 stimulation. Moreover, we also performed additional confocal experiments. As

shown in the new Fig. 3F and described in page 10, colocalization of gp130 and VPS26 following IL-6 stimulation was enhanced following IL-6 stimulation, indicating that gp130 underwent endocytosis and migrated to the retromer following IL-6 stimulation. Additionally, our original data showed that the associations of VPS26 with JAK1 and STAT3 were also enhanced upon IL-6 stimulation (Fig. 4C), and knockdown of VPS5 impaired the associations of JAK1 with gp130, STAT3 and VPS35 (Fig. 4A). Furthermore, knockdown of retromer-related proteins VPS35, VPS26 and VPS29 impaired IL-6-induced STAT3 activation (Fig. 3G). Collectively, these results suggest that the signaling complex consisting of gp130, JAK1 and STAT3 is recruited to the retromer-positive structures following IL-6 stimulation, and retromer can serve as platforms or “relay stations” for IL-6-induced STAT3 activation.

Thirdly, our original results showed the associations of retromer-localized protein TRIM27 with gp130, JAK1 and STAT3 were enhanced following IL-6 stimulation (Fig. 2D). Most importantly, knockdown of TRIM27 impaired the associations of gp130, JAK1 and STAT3 with VPS35 and VPS26 upon IL-6 stimulation (Fig. 4B&C), indicating that TRIM27 was essential for recruitment of signaling complex to retromer-positive structures. Moreover, overexpression of TRIM27 enhanced the association of JAK1 and STAT3, whereas knockdown of TRIM27 had the opposite effects (Fig. 2F&G), suggesting TRIM27 was required for mediating JAK1-STAT3 interaction.

In conclusion, our original and additional results indicate that TRIM27 acts as an essential adaptor for recruitment of gp130/JAK1/STAT3 signaling complexes to the retromer, which serves as a signaling platform or “relay station” for STAT3 activation. In addition, our results also suggest that TRIM27 is essential for the interaction of JAK1 and STAT3 at the retromer.

2. Fig. 1B: It has been previously reported that TRIM27 negatively regulates IKK α/β -mediated NF- κ B signals and IKK ϵ -mediated IRF signals. However, the authors showed that TRIM27 activated IL-6-mediated signals. Did the authors check several signal pathways other than the signals through IL-6 or IFN- β ? The authors should mention the molecular insights for the difference among these signals in the discussion section.

Reply:

We have examined the effects of TRIM27 on another IL-6 family cytokine OSM-induced STAT3 activation in our original manuscript. As shown in Figure S3, TRIM27 significantly potentiated OSM-induced STAT3 activation, whereas knockdown or knockout of TRIM27 had the opposite effects. We have now performed additional reporter assays to test the effects of TRIM27 on IFN- γ -induced STAT1 activation. As shown in the new Fig. S1A and described in page 6, overexpression of TRIM27 marginally enhanced IFN- γ -induced activation of IRF1 reporter (Fig. S1A). We have now discussed the difference on molecular mechanisms of these signal pathways in page 16 of the revised manuscript. It is possible that the distinct cellular localization, post-translational modification or complex formation of TRIM27 is responsible for its differential functions.

3. Fig. 1G: anti-TRIM27 immunoblots is poor or unreliable. The authors should clearly show anti-TRIM27 immunoblots. The bands were scraped in this manuscript.

Reply:

As the reviewer's suggested, we have now provided more clear images of anti-TRIM27 immunoblots in the new Fig. 1H (the original Fig. 1G) in the revised manuscript.

Reviewer #2 (Remarks to the Author):

.....

Data are obtained using ingenious strategies and supported by various model systems. In general, evidence provided is strong and supports the conclusions. Overall, the manuscript is clearly written.

Points that should be addressed are the following:

- In the title, and in many instances throughout the manuscript, it is stated that "TRIM27 mediates STAT3 activation at the retromer..." The authors should specify it either as 'retromer-positive structures' or 'compartment containing retromer'. Thus, stating in line 337 that "TRIM27 was localized at the retromer but not endosomes" is not accurate, since they meant localization to a retromer-positive compartment as opposite to early endosomes.

Reply:

Following the reviewer's suggestion, we have now specified it as "retromer-positive structures" in the title and other places in the manuscript, and have rewritten the sentence to "TRIM27 was localized at retromer-positive structures" in page 18 in the revised manuscript.

- Along these lines, when retromer is defined on line 107 as "a highly conserved heteropentameric complex important for recycling proteins from the endosomes..." nowhere in the manuscript the complex is properly defined. This would be appropriate in the introduction as well as in the discussion. Indeed, the Vacuolar protein sorting (Vps) complex Vps26/Vps35/Vps29 is a trimer considered – at least till recently – as the main cargo-selective subcomplex of retromer, yet the referred heteropentamer includes an interacting sorting nexin dimer, whose potential role is not addressed here.

Reply:

Following the review's suggestion, we have now defined the retromer in page 9 in the revised manuscript as "The mammalian retromer complex comprises a core cargo recognition trimer composed of VPS26, VPS29 and VPS35, and a sorting nexin heterodimer composed of SNX1 or SNX2 with SNX5, SNX6 or SNX27". As there are a number of sorting nexins (SNXs), at least including SNX1, SNX2, SNX5, SNX6 and SNX27, which associate with the core complex (VPS35/VPS29/VPS26) of

retromer, we have currently not investigated their functions in this study.

- Lines 222 & 345: “retromer acts as a platform for TRIM27-mediated assembly of JAK1-STAT3 complexes and subsequent activation of STAT3 following IL-6 stimulation.” The proposed role of retromer “as a platform” is recurrent throughout the text. Could perhaps the evidence provided indicate that retromer functions here in recruitment for sorting of signaling molecules or simple defines a location where the signaling molecules act? In other words, is it actively participating in TRIM27-mediated delivery of pSTAT3 toward the nucleus?

Reply:

As shown in Fig. 3 and Fig. 4, the associations of retromer components VPS35 and VPS26 with gp130, JAK1 and STAT3 were enhanced following IL-6 stimulation, indicating the signaling complex is recruited to the retromer-positive structures. In addition, knockdown of VPS35 impaired the associations of JAK1 with gp130, STAT3 and TRIM27 (Figure 4A). Moreover, knockdown of TRIM27 impaired the associations of JAK1 with STAT3, VPS26 and VPS35, as well as the associations of VPS26 with gp130, JAK1 and STAT3 (Fig. 4B&C). These results indicate the retromer-positive structure can serve as a location where the signaling molecules act. Therefore, the retromer-positive structure acts as a platform for TRIM27-mediated assembly of JAK1-STAT3 complexes and subsequent phosphorylation of STAT3 following IL-6 stimulation. It is currently unclear whether TRIM27 is involved in the translocation of pSTAT3 to the nucleus. We have modified our statement as “retromer acts as a platform for TRIM27-mediated assembly of JAK1-STAT3 complexes and subsequent phosphorylation of STAT3 following IL-6 stimulation” in page 11 and 18.

- The immunofluorescence data are somewhat poor, in part because images shown are at insufficient magnification; perhaps an overlap of endogenous Vps26 with TRIM27 and especially with pSTAT3 seen at higher magnification would help; there should be commercially available antibodies against Vps26 suitable to use on cells.

Reply:

Following the reviewer’s suggestion, the magnified immunofluorescence images have now been provided in new Fig. 2 and Fig. 3. We have also purchased various antibodies against VPS26, VPS35, pSTAT3 and TRIM27 and attempted to perform endogenous immunofluorescence experiments. Unfortunately, the antibodies were not good enough for endogenous immunofluorescence assays.

- Where cells in Fig 1A, B & C treated with IL6 for 10 h? This sounds in striking contrast with the 0-60 min treatment indicated for instance in Fig 1D.

Reply:

We are sorry for having not described the experiments in full details. For reporter assays in Fig 1A&B, the cells were treated with IL6 for 10 h. For qPCR analysis of IL6-induced transcription, the cells were treated with IL6 for 1 h. This has now been clearly described in the figure legends.

- Please ensure to indicate the cell model used in all figures and/or the corresponding legends. For instance, are data in Fig 1E & G obtained in HeLa cells as Fig 1F? Fig S3 legend appears also misleading.

Reply:

Following the reviewer's suggestion, the cell models used in the experiments have been described in detail in the corresponding figure legends. The data in new Fig. 1E, 1G and 1H (Fig. 1F&G in the original manuscript) were all obtained in HeLa cells. We are sorry for the mistake in the legend of the original Fig. S3 (new Fig. S4B-D). We have now corrected it.

- Blots display changes that are often not striking, and quantitation is unnecessary. Yet, please state in legends that a representative experiment is shown.

Reply:

Following the review's suggestion, the previously prepared samples were subjected to immunoblotting analysis with more concentrated antibodies against VPS35, TRIM27 and gp130. The more clear blots have been provided in the new Fig. 4A and Fig. 4C in the revised manuscript. As the reviewer suggested, we have stated in the legends that a representative experiment is shown or data are representative of two or three experiments with similar results.

- Line 81, please rewrite "into lipid vesicles called endosomes"; why not simply 'cellular vesicles'?

Reply:

We have corrected it following the reviewer's suggestion (see page 4 in the revised manuscript).

- A few typo errors should be fixed; e.g. please replace in line 103 "reported to involve in" for "involved in". Verb tenses should also be double checked.

Reply:

We have now corrected the typos.

- Line 198, and in other locations: please replace "punctuate" for "punctate".

Reply:

We have replaced "punctuate" for "punctate" in page 9 and other locations in the revised manuscript.

- Line 198: please take off "or microsomes", since this term refers to ER-derived artefactual structures obtained by cell fractionation.

Reply:

Following the reviewer's suggestion, "or microsomes" has been taken off in line page 9 in the revised manuscript.

- Line 207: "knockdown of VPS35 prolonged IFN-alpha-induced STAT1 phosphorylation (Fig 3E), which is consistent with a previous report (Ref 34)." Yes,

but keep in mind that in Ref 34–Fig 7A & B the effect (increase of approx. 50 %) is seen only at 60 min from induction, not before that.

Reply:

As shown in Fig. 3G, our results indicated that knockdown of VPS35 prolonged IFN- α -induced STAT1 phosphorylation at 30 min, but not at 15 min, whereas in the previous report the effect is seen at 60 min from induction. Although there existed a discrepancy in the effect time, the tendency that the effect of VPS35 knockdown on prolonging STAT1 phosphorylation induced by IFN- α was more prominent at later time course was consistent. The discrepancy in the effect time might result from the different cells used in the experiments (HeLa cells in our study vs RPE1 in Ref. 34), the sensitivity of cells responding to IFN- α , and the dose of IFN- α (20 ng/ml in our study vs 10000 U/ml in Ref. 34) used to stimulate cells.

- Line 217: The sentence: “These data suggest that the...” may be removed due to redundancy with that at the end of the paragraph.

Reply:

As the reviewer suggested, this sentence has been removed.

- Fig 5: Why is there such a high discrepancy in controls if the same experimental conditions were used? Compare A with C for HeLa cells (colony # goes from 50 to almost 200), and B with D for the mouse model (similarly, for tumor growth).

Reply:

We are sorry for having not precisely described the experiments in the legends of the original Fig. 5 (now new Fig. S5A-D in the revised manuscript). In these experiments, different cell densities were used. In the soft agar assays, TRIM27-overexpressing and corresponding control cells were seeded at 1000 per well, while TRIM27-RNAi and corresponding control cells were seeded at 2000 per well. In tumor xenograft experiments, a total of 10^6 TRIM27-overexpressed or corresponding control cells were injected into each mouse, while 2×10^6 TRIM27-RNAi or corresponding control cells were injected into each mouse.

- Fig. 6E & 7D: are 1, 2, 3 triplicates?

Reply:

The numbers “1, 2, 3” in the original Fig. 6E (new Fig. 6F) and 7D represent the colonic lysates collected from three independent mice. We have now added “#” ahead of the numbers and have described it in the legend.

- Fig 7D: a blot for Cyclin D1 is shown, however not mentioned in the text.

Reply:

The blot for Cyclin D1 was shown in original Fig. 7D to illustrate that deficiency of Trim27 also inhibited the expression of Cyclin D during tumor development. We have now mentioned it in page 14 and 17 in the revised manuscript.

- Line 339: “...recruitment of STAT3 to JAK1 as well as gp130, JAK1 and STAT3 to

the retromer but not the assembly of the retromer, suggesting that TRIM27 acts as a linker for recruiting gp130, JAK1 and STAT3 to the retromer.” (See also line 41 in the Abstract.) This sentence is confusing and should be rephrased. In particular, the interaction with gp130 is not shown in Fig 4. According to the data presented, retromer does not interact with gp130 (Fig 3D), whereas TRIM27 does it (Fig 2D).

Reply:

It is right that the retromer does not interact with gp130 in overexpression coimmunoprecipitation assays without IL-6 stimulation as shown in Fig. 3D. We have performed additional coimmunoprecipitation experiments and demonstrated that the retromer components VPS35 and VPS26 interacted with gp130 following IL-6 stimulation as shown in new Fig. 3E and Fig. 4C. We have also performed additional confocal experiments to detect whether gp130 colocalizes with VPS26. As shown in new Fig. 3F and described in page 10, gp130 colocalized with VPS26 and TRIM27 following IL-6 stimulation. These results indicate the association of gp130 with the retromer is induced by IL-6 stimulation.

- Line 470: the formula used to calculate tumor volume appears convenient, yet may not be accurate since most tumors have hemisphere (or ellipsoid) shape; $W = H$ according to the formula used. Why was not used instead $V = (\pi \text{ number}/6) \times L \times W \times H$ (or expressed differently, $V = (4/3) \pi \times r1 \times r2 \times r3$).

Reply:

The formula $V=1/2 \times L \times W^2$ used to calculate tumor volume in our original manuscript was derived from the formula $V = (\pi/6) \times L \times W \times H$ by assuming $\pi=3$ and $W=H^{2,3}$. As it is not easy to measure the height of tumor in living animals, the former formula was also used by many researchers. As the reviewer suggested, we have used the formula $V = (\pi/6) \times L \times W \times H$ to calculate volumes of tumors formed by RKO cells in the newly added Fig. 5 in the revised manuscript.

- Line 553: please add pg. numbers in Ref. 25.

Reply:

We have now added the page numbers of Ref. 25 in the revised manuscript.

- Line 898: please replace “as schematic shown” for “as in the schematic diagram shown” or “as in the schema shown”.

Reply:

We have now replaced “as schematic shown” for “as in the schematic diagram shown” in page 52 in the revised manuscript.

- Line 906: “minutes” should refer to “min”.

Reply:

We have now replaced “minutes” for “min” in page 52 of the revised manuscript.

Reviewer #3:

In this work, Zhang et al identify TRIM27 as a positive regulator of IL-6-induced STAT3 activation. Their data show that TRIM27 is required for JAK1-STAT3 complex formation. In addition, their data show that deletion of TRIM27 reduces DSS-induced acute inflammation and AOM/DSS-induced tumorigenesis. Although this study is interesting, some of key conclusions are not fully supported by the present experiments as outlined in more detail below.

Major points:

1. Most in vitro experiments were done in one cell line, HEK293. The key experiments should be performed in more than one cell line. Since the authors showed the function of TRIM27 in acute colonic inflammation and colitis-associated tumorigenesis, they should confirm their in vitro results in colon epithelial cells and/or colonic stroma cells.

Reply:

Two cell lines, HEK293 and HeLa, were mainly used to perform *in vitro* experiments in our original manuscript. As the reviewer suggested, we have now confirmed our *in vitro* results in two additional colonic epithelial cell lines, RKO and HT29, respectively. We constructed stable TRIM27-overexpressing and TRIM27-RNAi RKO cell lines and performed qPCR assays, immunoblots and reporter assays. As shown in new Fig. S1B-F, overexpression of TRIM27 potentiated IL-6 induced transcription of STAT3 target genes *SOCS3* and *FOS* as well as phosphorylation of STAT3 at Y705, while knockdown of TRIM27 had the opposite effects. We have mentioned these results in page 6 of the revised manuscript. We also generated stable TRIM27-RNAi cell lines to investigate the functions of endogenous TRIM27 in HT29 cells. As shown in new Fig. S1G&H, knockdown of TRIM27 inhibited IL-6-induced transcription of STAT3 target genes *SOCS3* and *FOS* as well as phosphorylation of STAT3 at Y705. Additionally, we have also subcutaneously injected either TRIM27-overexpressing or TRIM27-RNAi RKO cell lines together with the respective control cell lines into the flanks of nude mice to examine the roles of TRIM27 in tumor growth. As shown in the new Fig. 5 and described in page 11, overexpression of TRIM27 promoted anchorage-independent growth of RKO cells *in vitro* and tumor growth in nude mice, whereas knockdown had the opposite effects. These results demonstrate that TRIM27 also functions in colon epithelial cell lines.

2. Although the authors present data showing that IL-6 fails to induce activation of STAT3 and transcription of its downstream genes in TRIM27-deficient BMDMs (Fig. 1H-I) and in TRIM27-deficient primary hepatocytes, it is not clear whether IL-6 also fails to induce activation of STAT3 and transcription of its downstream genes in colonic epithelial cells and/or stroma cells.

Reply:

We have isolated the intestinal epithelial cells (IECs) from colons of WT and TRIM27-deficient mice, and stimulated with IL-6. As shown in the new Fig. 1J&K and described in pages 6-7, deficiency of TRIM27 inhibited IL-6-induced STAT3

phosphorylation at Y705 as well as the transcription of its downstream target genes *Socs3* and *Il6* in IECs (see new Fig. 1K&L). Our results indicate that TRIM27 functions in divergent types of cells.

3. DSS-treated and DSS/AOM-treated mice are not good models for IBD and IBD-associated CRC because the DSS model represents a process of injury and wound healing. Therefore, the role of TRIM27 in IBD and IBD-associated CRC should be evaluated in other mouse models of IBD and IBD-associated CRC.

Reply:

Although mouse models of DSS-induced colitis and AOM/DSS-induced colitis-associated cancer (CAC) development cannot fully represent the complexity of IBD and IBD-associated CRC development in humans, they are useful in providing insights into the mechanisms about inflammation-associated cancer development and are widely used in the field. Oral administration of DSS to mice via drinking water induces severe colitis characterized by weight loss, bloody diarrhea, ulcer formation, loss of epithelial cells and infiltrations with neutrophils, resembling certain features of human ulcerative colitis⁴. In addition, it has been demonstrated that the IL-6-STAT3 signaling pathways play important roles in CAC development^{5, 6, 7, 8}. As TRIM27 is involved in regulation of IL-6-induced STAT3 activation, we believe that it is appropriate to use these models to study the functions of TRIM27 in CAC development.

4. It seems that the results from Fig. 2A-B indicate that TRIM27 can replace TYK2 in IL-6-induced STAT3 activation. Is this correct? Does TRIM27 interact with TYK2?

Reply:

JAK kinases are required for IL-6-induced STAT3 activation by mediating STAT3 phosphorylation at Y705. The results of the new Fig. S4A and Fig. 2A (Fig. 2A&B in the original manuscript) that the dominant negative mutant of TYK2 had no effects on TRIM27-mediated STAT3 activation indicated that TRIM27 did not target TYK2 in regulation of STAT3 activation, and thus TYK2 was not required for TRIM27-mediated STAT3 activation and not taken into further consideration in our study. We currently do not know whether TRIM27 interacts with TYK2.

5. In Fig. 2D, does IL-6 really induce the association of TRIM27 with gp130 and JAK2. Similarly, it's hard to see how IL-6-induced the association of JAK1 with TRIM27 and VPS35 in Fig. 4A. much better images should be provided.

Reply:

As the reviewer suggested, we performed additional immunoblotting analysis to analyze the previously prepared samples. We optimized the blotting conditions and obtained more clear blots, which have now been shown in the new Fig. 2C (Fig. 2D in the original manuscript) and Fig. 4A. The results indicated that the association of TRIM27 with gp130 was indeed enhanced upon IL-6 stimulation (new Fig. 2C), whereas the association of TRIM27 and JAK2 was quite weak and not induced by IL-6 stimulation. The blots of JAK2 were removed from the new Fig. 2C. In addition,

the associations of JAK1 with TRIM27 and VPS35 were also increased following IL-6 stimulation as shown in new Fig. 4A.

6. *Does knockdown of TRIM27 reduce the association of VPS35 with JAK1 and STAT3?*

Reply:

As shown in Fig. 4B, knockdown of TRIM27 reduced the association of VPS35 and JAK1. Although there was no direct evidence, it was supposed that the association of VPS35 with STAT3 would be reduced when TRIM27 was knockdown, since knockdown of TRIM27 impaired the association of STAT3 and VPS26 (Fig. 4C), which formed complex with VPS35.

7. *Since the error bars have wide variations in Fig. 7B, it's necessary to confirm that the difference of tumor numbers between WT and TRIM27-deficient mice is significant.*

Reply:

We have repeated the statistical analysis using larger-sized samples (n=16). The sample size in the original manuscript was n=8. As shown in the new Fig. 8B (Fig. 7B in the original manuscript), the difference of tumor numbers in total (p=0.000005), tumor numbers of size of 2-4 mm (p=0.00009) and tumor numbers of size of >4 mm (p=0.01028) between WT and TRIM27-deficient mice is significant, while tumor numbers of size of <2 mm (p=0.088) is insignificant. The conclusion is the same.

8. *Does loss of TRIM27 reduce colonic inflammation in AOM/DSS-treated mice?*

Reply:

The body weight of WT and TRIM27-deficient mice were monitored during the process of tumor induction by AOM/DSS treatment. As shown in Fig S6A and described in page 14, the body weight changes between WT and TRIM27-deficient mice during AOM/DSS treatment were not as notable as that of during the process of DSS-induced acute colitis. In addition, we have also performed additional ELISA assays to detect inflammatory cytokines in sera and colon tissues at day 14 and day 36. As shown in Fig. S6B and mentioned in pages 17-18, the amounts of inflammatory cytokines IL-6 in sera and colon tissues of WT mice were only slightly higher than that of TRIM27-deficient mice at body day 14 and day 36. The amounts of TNF- α in sera at day 14 and colon tissues at day 36 of WT mice were also slightly higher than that of TRIM27-deficient mice, while there were no difference in the amounts of TNF- α in sera at day 36 and colon tissues at day 14 between two group mice. These results suggest that loss of TRIM27 marginally reduces colonic inflammation in AOM/DSS-treated mice.

Minor points:

1. *Dose knockdown of TRIM27 also inhibit IL-6-induced STAT3 transactivation (STAT3 report activity)?*

Reply:

We have performed additional reporter assays to examine the effects of TRIM27 knockdown on IL-6-induced STAT3 transactivation. As shown in the new Fig. 1F and Fig. S1F, knockdown of TRIM27 inhibited IL-6-induced activation of STAT3 reporter in both HeLa cells and RKO cells. These results indicate knockdown of TRIM27 also inhibits IL-6-induced STAT3 transactivation. We have included these results in page 6 in the revised manuscript.

2. *Does overexpression of TRIM27 in HEK293 promote cell anchorage-independent growth and tumor growth in nude mice?*

Reply:

To answer the reviewer's question, we generated stable TRIM27-overexpressing or TRIM27-RNAi HEK293 cell lines and performed additional soft agar and tumor xenografts experiments. As shown in the new Fig. S5E&F and described in page 11, overexpression of TRIM27 in HEK293 promoted cell anchorage-independent growth, whereas knockdown of TRIM27 had the opposite effects. We have also subcutaneously injected either TRIM27-overexpressing or TRIM27-RNAi HEK293 cell lines together with the respective control cell lines into the flanks of nude mice to examine the role of TRIM27 in tumor growth, but the tumor did not grow perhaps due to the low tumorigenicity of HEK293 cells. Thus, a colonic epithelial cell line, RKO, was used to further study the roles of TRIM27 in tumorigenesis. As shown in the new Fig. 5 and described in page 11, overexpression of TRIM27 in RKO cells promoted tumorigenesis, while knockdown of TRIM27 had the opposite effects. These results indicate TRIM27 plays important role in tumorigenesis.

Reviewer #4:

.....

While the observations are novel and interesting, three key questions must be addressed:

1) What cell type is crucial for STAT3 activation via TRIM27 in acute DSS colitis? STAT3 plays cell type-dependent roles in the inflamed gastrointestinal tract. Is TRIM27-induced STAT3 activation myeloid-specific or IEC-specific? STAT3 in IEC maintains barrier integrity and survival, while STAT3 in myeloid cells may mediate anti-inflammatory effects and promote formation of colitis-associated tumors.

Reply:

To determine the cell types that are crucial for STAT3 activation via TRIM27 in DSS-induced acute colitis, we have generated reciprocal bone marrow chimeric mice by adoptively transferring bone marrow to lethally irradiated recipients. Eight weeks after bone marrow transplantation, these mice were subjected to DSS challenge. The results were included as new Fig. 7 and described in page 13. As expected, TRIM27-deficient mice transplanted with hematopoietic cells from TRIM27-deficient mice (KO>KO) were less susceptible to DSS-induced colitis than wild-type mice receiving hematopoietic cells from wild-type mice (WT>WT) as determined by less

body weight loss, higher survival rate, reduced colon shortening, less inflammatory cells infiltration, lower histological score, decreased level of phosphorylated STAT3 and p65 and lower expression of inflammatory cytokines (see new Figure 7A-H). However, TRIM27-deficient mice transplanted with hematopoietic cells from wild-type mice (WT>KO) developed colitis as severe as that of WT>WT mice (new Figure 7A-H). Conversely, wild-type mice transplanted with hematopoietic cells from TRIM27-deficient mice (KO>WT) showed decreased susceptibility to colitis as KO>KO mice (new Figure 7A-H). These results indicate TRIM27-deficient hematopoietic cells are responsible for attenuation of DSS-induced colitis.

In addition, we have also isolated the IECs from colons of WT and TRIM27-deficient mice, and stimulated with IL-6. As shown in new Fig. 1K&L, deficiency of TRIM27 inhibited IL-6-induced transcription of its downstream target genes as well as phosphorylation of STAT3 at Y705 in IECs. Collectively, these results indicate TRIM27 functions both in hematopoietic cells and IEC cells in regulation of STAT3 activation.

2) TRIM27 has previously been shown to act as a negative regulator of type I IFN production. Signaling through type I IFN receptors may regulate the activation of STAT3. In animal models of IBD it has been shown that type I IFN may protect against colitis. Is it possible that the actual link between TRIM27 and STAT3 is INDIRECTLY through altered type I IFN production - at least in vivo?

Reply:

We have performed additional ELISA assays to detect the amounts of type I IFNs (IFN- α and IFN- β) produced in sera and colonic mucosa of WT and Trim27-deficient mice at day 14 and day 36 during the course of AOM/DSS treatment. As shown in new Fig. S6C and described in page 18, the amounts of type I IFNs were relatively low and there were no any difference in type I IFNs production in colon tissues between TRIM27-deficient mice and its wild-type littermates in our mouse model. In addition, we could not detect any type I IFNs in the sera of TRIM27-deficient mice and their wild-type littermates. Therefore, the actual link between TRIM27 and STAT3 seems to be direct.

3) Did TRIM27^{-/-} develop fewer tumors because they just developed less mucosal inflammation in this model?

Reply:

Our results indicated TRIM27 could function both in hematopoietic cells and IECs to regulate STAT3 activation (please see our reply to Specific Point #1). In addition, as shown in Fig. 8D, the expression of STAT3 downstream target genes *Bcl-xl*, *c-Myc*, *Pcna* and *Ccnd1*, which were responsible for tumor cell survival and proliferation respectively, were dramatically down-regulated in colon tumors of TRIM27-deficient mice compared to their wild-type littermates. Therefore, less mucosal inflammation orchestrated by hematopoietic cells and impaired activation of STAT3 as well as expression of pro-oncogenic STAT3 target genes in IECs are supposed to be main factors responsible for the development of fewer tumors in

TRIM27-deficient mice in this model.

Minor:

1) *Histology score is missing in Figure 6.*

Reply:

We have now evaluated histological scores as described⁹ and included two panels (new Fig. 6E, Fig. 7E) of histological score in the revised manuscript. As shown in new Fig. 6E, the histological score of WT mice group is higher than that of TRIM27-deficient mice group, indicating colitis in WT mice group is more severe than that of TRIM27-deficient mice group. As shown in new Fig. 7E, the histological scores of mice groups of WT>WT and WT>KO are higher than those of mice groups of KO>WT and KO>KO, indicating colitis in mice groups of WT>WT and WT>KO is more severe than that of KO>WT and KO>KO.

2) *In each figure/panel, it must be clearly indicated how many mice per group were investigated per experiment/assay.*

Reply:

We have now done this as the reviewer suggested.

3) *Please indicate in MM whether mice were age/gender-matched and whether mice were specific pathogen-free. Viral co-triggers have been shown to influence acute DSS colitis in animal facilities. TRIM27 may play a role in virus-host interactions – therefore please indicate in MM that mice were specifically virus-free.*

Reply:

Yes, the mice were age/gender-matched and maintained in SPF facilities. We have now added the information in the revised manuscript.

References

1. Macia E, Ehrlich M, Massol R, Boucrot E, Brunner C, Kirchhausen T. Dynasore, a cell-permeable inhibitor of dynamin. *Dev Cell* **10**, 839-850 (2006).
2. Euhus DM, Hudd C, LaRegina MC, Johnson FE. Tumor measurement in the nude mouse. *J Surg Oncol* **31**, 229-234 (1986).
3. Tomayko MM, Reynolds CP. Determination of subcutaneous tumor size in athymic (nude) mice. *Cancer Chemother Pharmacol* **24**, 148-154 (1989).
4. Wirtz S, *et al.* Chemically induced mouse models of acute and chronic intestinal inflammation. *Nat Protoc* **12**, 1295-1309 (2017).
5. Greten FR, *et al.* IKKbeta links inflammation and tumorigenesis in a mouse model of colitis-associated cancer. *Cell* **118**, 285-296 (2004).
6. Grivennikov S, *et al.* IL-6 and Stat3 are required for survival of intestinal epithelial cells and development of colitis-associated cancer. *Cancer Cell* **15**, 103-113 (2009).
7. Bollrath J, *et al.* gp130-mediated Stat3 activation in enterocytes regulates cell survival and cell-cycle progression during colitis-associated tumorigenesis.

- Cancer Cell* **15**, 91-102 (2009).
8. Bromberg J, Wang TC. Inflammation and cancer: IL-6 and STAT3 complete the link. *Cancer Cell* **15**, 79-80 (2009).
 9. Erben U, *et al.* A guide to histomorphological evaluation of intestinal inflammation in mouse models. *Int J Clin Exp Pathol* **7**, 4557-4576 (2014).

REVIEWERS' COMMENTS:

Reviewer #1 (Remarks to the Author):

It is now acceptable for publication in Nature Communications.

Reviewer #2 (Remarks to the Author):

The authors performed a considerable amount of work that addressed the reviewers' comments, demonstrating that TRIM27 mediates IL6-induced STAT3 activation.

A few minor points:

- Figure S4. It's not exclusively related to Figure 2, since also shows that activation "is dependent on IL-6Ra/gp130 endocytosis" (panels E & F)
- Figure S6B, C. Should show chart legend for Trim27^{+/+} and Trim27^{-/-} mice
- Long sentence beginning in line 390 should be rewritten for better understanding: perhaps shortening by saying (lines 393-394) "...in IECs may lead to development of..." instead of "...in IECs might be main factors responsible for development of..."
- Line 401 till the end of the paragraph: since there's an obvious confusion between Vps35 and TRIM27 knockdowns, it's not yet clear and should be rewritten. A suggestion, according to what shown in Fig. 4, could be: (Line 401) "Retromer knockdown impaired association of STAT3, gp130 and TRIM27 with JAK1, and inhibited IL-6-induced transcription of downstream genes. Likewise, TRIM27 knockdown impaired association of STAT3, gp130 and JAK1 with retromer, but not retromer assembly. These results suggest that the retromer complex acts as a platform for TRIM27-mediated assembly of JAK1-STAT3 and subsequent phosphorylation of STAT3 following IL-6 stimulation."

Reviewer #3 (Remarks to the Author):

The authors have addressed all of my concerns and I feel that the manuscript is now ready for publication.

Reviewer #4 (Remarks to the Author):

The authors have addressed my 3 major concerns adequately.

In addition, I suggest the following recent reference in the discussion:

"TRIM27 functions as an oncogene by activating epithelial-mesenchymal transition and p-AKT in colorectal cancer."

<https://www.ncbi.nlm.nih.gov/pubmed/29767249>

Otherwise no further comments.

Point-by-point response

Reviewer #2:

A few minor points:

- *Figure S4. It's not exclusively related to Figure 2, since also shows that activation "is dependent on IL-6Ra/gp130 endocytosis" (panels E & F)*

Reply: We have now corrected it as "related to Figure 2 and Figure 3".

- *Figure S6B, C. Should show chart legend for Trim27^{+/+} and Trim27^{-/-} mice.*

Reply: We have now added the chart legends for Trim27^{+/+} and Trim27^{-/-} mice.

- *Long sentence beginning in line 390 should be rewritten for better understanding: perhaps shortening by saying (lines 393-394) "...in IECs may lead to development of..." instead of "...in IECs might be main factors responsible for development of..."*

Reply: We have now rewritten the sentence as the reviewer suggested.

- *Line 401 till the end of the paragraph: since there's an obvious confusion between Vps35 and TRIM27 knockdowns, it's not yet clear and should be rewritten. A suggestion, according to what shown in Fig. 4, could be: (Line 401) "Retromer knockdown impaired association of STAT3, gp130 and TRIM27 with JAK1, and inhibited IL-6-induced transcription of downstream genes. Likewise, TRIM27 knockdown impaired association of STAT3, gp130 and JAK1 with retromer, but not retromer assembly. These results suggest that the retromer complex acts as a platform for TRIM27-mediated assembly of JAK1-STAT3 and subsequent phosphorylation of STAT3 following IL-6 stimulation."*

Reply: We have now rewritten the sentence as the reviewer suggested.

Reviewer #4:

The authors have addressed my 3 major concerns adequately.

In addition, I suggest the following recent reference in the discussion:

"TRIM27 functions as an oncogene by activating epithelial-mesenchymal transition and p-AKT in colorectal cancer."

<https://www.ncbi.nlm.nih.gov/pubmed/29767249>

Otherwise no further comments.

Reply: We have now added the reference in the discussion.